

Atmospheric
Measurement
Techniques

# Retrieval of aerosol properties from Airborne Hyper-Angular Rainbow Polarimeter (AirHARP) observations during ACEPOL 2017

**Anin Puthukkudy**[1,2,3]**, J. Vanderlei Martins**[1,2,3]**, Lorraine A. Remer**[2,3]**, Xiaoguang Xu**[2,3]**, Oleg Dubovik**[4],
**Pavel Litvinov**[5]**, Brent McBride**[1,2,3]**, Sharon Burton**[6]**, and Henrique M. J. Barbosa**[1,7]

[1]Physics Department, University of Maryland Baltimore County, Baltimore, MD 21250, USA
[2]Joint Center for Earth Systems Technology, University of Maryland Baltimore County,
5523 Research Park DR, Baltimore, MD 21228, USA
[3]Earth and Space Institute, University of Maryland Baltimore County, Baltimore, 3Earth and Space Institute, University of Maryland Baltimore County, MD 21250 TS1 , USA
[4]Laboratoire d'Optique Atmosphérique, CNRS/Université Lille, Villeneuve-d'Ascq, France
[5]GRASP-SAS, Villeneuve-d'Ascq, France
[6]NASA Langley Research Center, Hampton, VA 23681, USA
[7]Instituto de Física, Universidade de São Paulo, São Paulo, Brazil

**Correspondence:** Anin Puthukkudy (anin1@umbc.edu)

**Abstract.** Multi-angle polarimetric (MAP) imaging of Earth scenes can be used for the retrieval of microphysical and optical parameters of aerosols and clouds. The Airborne Hyper-Angular Rainbow Polarimeter (AirHARP) is an aircraft MAP instrument with a hyper-angular imaging capability of 60 along-track viewing angles at 670 nm and 20 along-track viewing angles at other wavelengths – 440, 550, and 870 nm – across the full 114° (94°) along-track (cross-track) field of view. Here we report the retrieval of aerosol properties using the Generalized Retrieval of Aerosols and Surface Properties (GRASP) algorithm applied to AirHARP observations collected during the NASA Aerosol Characterization from Polarimeter and Lidar (ACEPOL) campaign in October–November 2017. The retrieved aerosol properties include spherical fraction (SF), aerosol column concentration in multiple size distribution modes, and, with sufficient aerosol loading, complex aerosol refractive index. From these primary retrievals, we derive aerosol optical depth (AOD), Angstrom exponent (AE), and single scattering albedo (SSA). AODs retrieved from AirHARP measurements are compared with the High Spectral Resolution LiDAR-2 (HSRL2) AOD measurements at 532 nm and validated with measurements from collocated Aerosol Robotic NETwork (AERONET) stations. A good agreement with HSRL2 ($\rho = 0.940$, $|BIAS| = 0.062$, mean absolute error (MAE) $= 0.122$) and AERONET AOD ($0.010 \leq MAE \leq 0.015$, $0.002 \leq |BIAS| \leq 0.009$) measurements is observed for the collocated points. There was a mismatch between the HSRL2- and AirHARP-retrieved AOD for the pixels close to the forest fire smoke source and to the edges of the plume due to spatial mismatch in the sampling. This resulted in a higher BIAS and MAE for the HSRL2 AOD comparison. For the case of AERONET AOD comparison, two different approaches are used in the GRASP retrievals, and the simplified aerosol component-based GRASP/Models kernel which retrieves fewer number of aerosol parameter performed well compared to a more generous GRASP/Five mode approach in the low aerosol loading cases. Forest fire smoke intercepted during ACEPOL provided a situation with homogenous plume and sufficient aerosol loading to retrieve the real part of the refractive index (RRI) of 1.55 and the imaginary part of the refractive index (IRI) of 0.024. The derived SSAs for this case are 0.87, 0.86, 0.84, and 0.81 at wavelengths of 440, 550, 670, and 870 nm, respectively. Finer particles with

an average AE of 1.53, a volume median radius of 0.157 μm, and a standard deviation (SD) of 0.55 for fine mode is observed for the same smoke plume. These results serve as a proxy for the scale and detail of aerosol retrievals that are anticipated from future space mission data, as HARP Cube-Sat (mission begins 2020) and HARP2 (aboard the NASA PACE mission with launch in 2023) are near duplicates of AirHARP and are expected to provide the same level of aerosol characterization.

# 1   Introduction

Aerosols play an important role in Earth's climate (Boucher et al., 2013; Hobbs, 1993; Kaufman et al., 2002; Koren et al., 2004): they directly perturb Earth's radiation budget and indirectly modify cloud properties, which in turn influences the planet's energy and hydrological budgets (Lenoble et al., 2010; Penner et al., 2001). The *direct radiative effects* of aerosols, the absorption and scattering of light, depend on the intrinsic optical properties of the particles, the total aerosol loading, and the radiative properties of the surface beneath the aerosol layer. Aerosols are highly variable, both in their spatial and temporal distributions, but also in their optical and microphysical properties; it is especially challenging to represent their radiative effect realistically in climate models (Dubovik et al., 2002; Masmoudi et al., 2003). Therefore, aerosol radiative forcing remains one of the main uncertainties in global climate change estimation (Boucher et al., 2013; Chen and Penner, 2005; Hansen et al., 2011; Penner et al., 2011). Furthermore, aerosols are a mixture of submillimeter, suspended particles with different sizes, morphology, and composition that result in complex physical, chemical, and optical properties (Kahnert, 2010; Kokhanovsky et al., 2015; Tanré et al., 2011). To better characterize the aerosol role in the global radiation budget and narrow uncertainties in predicting climate change, we need to better understand and constrain the temporal and spatial distributions of these properties. In addition, a careful understanding of aerosol properties is essential for air quality monitoring/mitigation, characterizing fertilization of ecosystems, hydrological forecasting, etc. (Shiraiwa et al., 2017; Westberry et al., 2019).

The last few decades have seen unprecedented efforts to better characterize aerosol particle properties and aerosol radiative effects with in situ and remote sensing observations. For example, in situ measurements based on a wide variety of techniques, such as photoacoustic and cavity ring-down spectrometers, filter-based photometers, and polarized imaging nephelometers, have provided detailed information on size, shape, and absorption for many different regions across the world and continue to do so (Bergstrom et al., 2007; Bond et al., 1999; Dubovik et al., 2000; Espinosa et al., 2017, 2018; Moosmüller et al., 2005; Petzold et al., 2005; Rocha-Lima

et al., 2014; Snider et al., 2015). However, in situ measurements have limitations due to the small sampling volumes that they represent and are very limited in number and spatial coverage. In addition to in situ instruments, ground-based remote sensing networks, primarily the AErosol RObotic NETwork (AERONET), provide much larger coverage over the continental Earth (Holben et al., 1998, 2001). These AERONET observations measure downwelling direct sunlight, and from these measurements, it is possible to obtain highly accurate spectral aerosol optical depth (AOD), defined as the integration of the aerosol extinction over the entire atmospheric column. In addition, AERONET instruments measure diffused and polarized sky radiance, from which columnar particle optical and microphysical properties are retrieved (Dubovik et al., 2000, 2006; Xu and Wang, 2015). AERONET instruments are widespread but are not truly global.

In order to achieve seamless global coverage, we need to rely on satellite remote sensing to characterize the global aerosol system, including particle properties. Most aerosol products retrieved from satellite instrument data are limited to AOD or qualitative aerosol type (Diner et al., 2008; Kahn et al., 2009; Lenoble et al., 2010; Levy et al., 2013; Limbacher and Kahn, 2019; Martonchik et al., 2002), whereas a multi-angle polarimeter (MAP) has enough information content to retrieve particle properties with a greater degree of accuracy (Dubovik et al., 2011; Hasekamp and Landgraf, 2007; Knobelspiesse et al., 2012; Mischenko et al., 2002; Mishchenko and Travis, 1997). A MAP instrument looks at Earth scenes at different viewing angles and measures the angular scattering and polarization of reflected light after interacting with Earth's surface, atmospheric molecules, clouds, and aerosols. Using multiple polarization angles and multiple wavelengths (if available), the aerosol signal can be isolated from the signals coming from the atmosphere and the surface beneath. Furthermore, these algorithms can invert MAP measurements to obtain optical properties of the aerosol within a significant level of certainty. This capability has been demonstrated by spaceborne POLDER I, II, and III (POLarization and Directionality of the Earth's Reflectance) (Deuzé et al., 1999, 2001; Dubovik et al., 2019; Goloub et al., 1999; Hasekamp et al., 2011; Leroy et al., 1997) and will be continued by the Multi-viewing multi-channel multi-polarisation imager (3MI): a future MAP instrument from the POLDER heritage scheduled to launch in 2021 (Fougnie et al., 2018). Currently, there are several modern MAP concepts that demonstrate technological advancements relative to the original POLDER, designed specifically as proxies for future spaceborne missions. These include, in addition to 3MI, Research Scanning Polarimeter (RSP) (Cairns et al., 1999, 2003), Airborne Multiangle Spectro Polarimetric Imager (AirMSPI) (Diner et al., 2013), SPEX Airborne (Smit et al., 2019), Observing System Including PolaRisation in the Solar Infrared Spectrum (OSIRIS) which is a 3MI airborne simulator, and Hyper-Angular Rainbow Polarimeter (HARP)

(Martins et al., 2018; Mcbride et al., 2020). There are several aerosol retrieval algorithms specifically optimized for MAPs, which include the SRON multi-mode inversion algorithm for SPEX airborne (Fu et al., 2020; Fu and Hasekamp, 2018); Microphysical Aerosol Property from Polarimeters (MAPP) (Stamnes et al., 2018) and GISS/RSP algorithm (Knobelspiesse et al., 2011; Waquet et al., 2009) for RSP; and correlated multi-pixel and joint retrieval algorithm for AirMSPI developed at Jet Propulsion Laboratory (JPL) (Xu et al., 2017, 2019). This list is not complete, and for a comprehensive review of the polarimetric remote sensing of atmospheric aerosols based on MAPs, we encourage the readers to refer to several reviews in the literature (Dubovik et al., 2019; Kokhanovsky et al., 2010, 2015; Remer et al., 2019). In this work, we focus on retrieval of aerosol properties using Airborne Hyper-Angular Rainbow Polarimeter (AirHARP) data, the airborne version of HARP, from the NASA aircraft campaign Aerosol Characterization from Polarimeter and Lidar (ACEPOL). We apply an inversion algorithm to AirHARP-polarized measurements of the same target at different viewing angles and wavelengths. The specific inversion algorithm is Generalized Retrieval of Aerosols and Surface Properties (GRASP) (Dubovik et al., 2011, 2014).

In Sect. 2, we provide a theoretical background for the measurements of multi-angle polarimetry and the inversion of those measurements to retrieve aerosol properties and then describe the AirHARP instrument specifically, define the measuring geometry, and introduce the specific campaign when the measurements were made. Section 3 describes the GRASP retrieval after covering the preliminary work preparing measurements for retrieval, including gas corrections. The results of applying GRASP to AirHARP measurements, including comparisons to collocated High Spectral Resolution LiDAR – 2 (HSRL2) and AERONET, are shown and discussed in Sects. 4 and 5. Section 6 discusses future research directions. Finally, Sect. 7 offers a conclusion. In addition, we provide two Appendices. One details the land and ocean surface models that are essential to the GRASP inversion of aerosol, and the second one describes the calculation of aerosol optical depth from retrieved aerosol particles.

## 2 Background

### 2.1 Theoretical basis of the measurements and retrieval

MAP instruments measure radiances at different viewing angles, polarization angles, and spectral bands. The state of polarization of measured light can be represented by the Stokes vector, $S$ TS2, where the transpose of the vector is given as $S^{\mathrm{T}} = [I \quad Q \quad U \quad V]$ (Schott, 2009). The elements in the Stokes vector are TS3

$$
\begin{bmatrix} I \\ Q \\ U \\ V \end{bmatrix} = \begin{bmatrix} E_\parallel E_\parallel + E_\perp E_\perp \\ E_\parallel E_\parallel^* - E_\perp E_\perp^* \\ E_\parallel E_\perp^* + E_\perp E_\parallel^* \\ i(E_\parallel E_\perp^* - E_\perp E_\parallel^*) \end{bmatrix},
\tag{1a}
$$

where $E_\perp$ and $E_\parallel$ are the perpendicular and parallel components of the electric field $E$, respectively. The first element ($I$) represents the total radiance. The second and third elements ($Q, U$) represent the linear polarization of the radiance, and the fourth element ($U$) represents circular polarization. Passive remote sensors, like AirHARP, use the Sun as their light source. Therefore, sunlight incident on the atmosphere is defined as

$$
S_{\mathrm{Inc}}^{\mathrm{T}} = \begin{bmatrix} I & 0 & 0 & 0 \end{bmatrix}.
\tag{1b}
$$

Since the light from the Sun is unpolarized, $Q, U$, and $V$ of the Stokes vector are zero. The Stokes vector of the scattered light back into the instrument sensor is given by

$$
S_{\mathrm{sca}}^{\mathrm{T}} = \begin{bmatrix} I_{\mathrm{sca}} & Q_{\mathrm{sca}} & U_{\mathrm{sca}} & 0 \end{bmatrix},
\tag{1c}
$$

where the light reaching the instrument sensor has now acquired some polarization but is assumed to be only linearly polarized, an assumption that holds well for the Earth's atmosphere and surface (Dubovik et al., 2011; Kokhanovsky et al., 2015). In this paper, we use reduced radiances $R_I = \frac{\pi I_{\mathrm{sca}}}{F_0}$, $R_Q = \frac{\pi Q_{\mathrm{sca}}}{F_0}$, and $R_U = \frac{\pi U_{\mathrm{sca}}}{F_0}$ to define the Stokes vector of the scattered light measured by the MAP with $R_I$, $R_Q$, and $R_U$ notation. $F_0$ is the solar radiance (Wm$^{-2}$ µm$^{-1}$), and hence $R_I$, $R_Q$, and $R_U$ are dimensionless variables. $R_I$ is the total radiation measured by MAP, the same that would be measured by a radiometer normalized by $F_0/\pi$. $R_Q$ and $R_U$ define orthogonal states of linear polarization and, together, they form the *polarized intensity*, defined as (Schott, 2009)

$$
R_{\mathrm{P}} = \sqrt{R_Q^2 + R_U^2},
\tag{1d}
$$

and the degree of linear polarization (DoLP) is

$$
\mathrm{DoLP} = R_{\mathrm{P}}/R_I.
\tag{1e}
$$

### 2.2 AirHARP (Airborne Hyper-Angular Rainbow Polarimeter)

The Hyper-Angular Rainbow Polarimeter (HARP) is a modern MAP concept capable of wide field-of-view (FOV), multi-angle, multi-wavelength polarimetric imagery of a ground scene even from a low-cost, CubeSat-size platform. The HARP program was initially funded by the NASA Earth Science and Technology Office (ESTO) InVEST program as a joint collaboration between the University of Maryland, Baltimore County (UMBC) in Baltimore, Maryland, and the Space Dynamics Laboratory at Utah State University in Logan, Utah. There are currently three instruments

https://doi.org/10.5194/amt-13-1-2020

Atmos. Meas. Tech., 13, 1–30, 2020

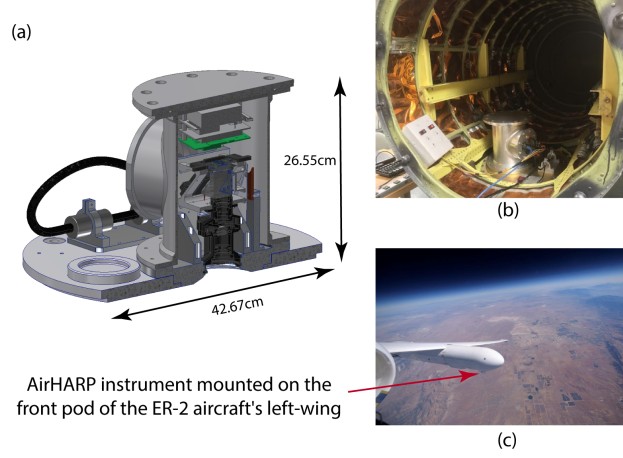

**Figure 1. (a)** Cross-section image of the AirHARP instrument rendered using CAD software; **(b)** AirHARP instrument mounted on the ER-2 aircraft left-wing front pod (image courtesy of Brent McBride); **(c)** image captured by NASA Science Pilot D. Stuart Broce during the ACEPOL flight on 26 October 2017. The red arrow in the figure points to the AirHARP's exposed part when mounted on the wing pod.

**Table 1.** Information on AirHARP spectral bands, viewing angles, and measured parameters.

| Band nominal wavelength | No. of viewing angles | Measured Variables |
| --- | --- | --- |
| 440 nm | 20 | I, Q, U |
| 550 nm | 20 | I, Q, U |
| 670 nm | 60 | I, Q, U |
| 870 nm | 20 | I, Q, U |

based on the original HARP concept: HARP CubeSat, a self-contained space technology demonstration mission launched to the International Space Station in November 2019 for a 1-year long mission beginning in February 2020; HARP2, a payload instrument for NASA's Plankton, Aerosols, Clouds, ocean Ecosystem (PACE) mission set to launch in the early 2023s (Werdell et al., 2019); and AirHARP, an airborne version of the HARP concept. In this paper, we focus on aerosol retrievals derived from measurements made from AirHARP as a proxy for these future space missions.

AirHARP's swath spans an angle of 114° along-track and 94° in cross-track; a simulated cross-section image of the AirHARP instrument is shown in Fig. 1a. It uses a Phillips prism system that splits the incoming beam of light into three components so that the radiation can be measured at three polarization angles simultaneously, with no moving parts. These polarization states are imaged on three CCD imaging sensors, denoted by A, B, and C, which measure the light at angles of linear polarization (AoLP) = 0°, 45°, and 90°, respectively, which are hereby denoted as $I_A$, $I_B$, and $I_C$ radiances. AirHARP has three wavelengths (440, 550,

and 870 nm) that measure at 20 along-track view angles plus a hyper-angle measurement at the 670 nm wavelength that measures at 60 along-track view angles (see Table 1). This capability allows AirHARP to view a single ground target from up to 60 different perspectives and measure the angular scattering response emanating from that location in both total and polarized radiances. These radiances are measured in all four channels, and each collocated detector pixel, which corresponds to a single channel and view angle, is calibrated independently for the radiometric and polarimetric measurements.

Using the calibration matrix **C** and measured $I_A$, $I_B$, and $I_C$, the $I_{sca}$, $Q_{sca}$, and $U_{sca}$ elements of the Stokes vector are calculated using Eq. (2) (Fernandez-Borda et al., 2009) and subsequently reduced radiances $R_I$, $R_Q$, and $R_U$ for each collocated detector pixel. TS4

$$\begin{bmatrix} I_{sca} \\ Q_{sca} \\ U_{sca} \end{bmatrix} = \begin{bmatrix} C_{11} & C_{12} & C_{13} \\ C_{21} & C_{22} & C_{23} \\ C_{31} & C_{32} & C_{33} \end{bmatrix} \begin{bmatrix} I_A \\ I_B \\ I_C \end{bmatrix} \quad (2)$$

The calibration matrix, **C**, is derived from a laboratory calibration scheme described in Fernandez-Borda et al. (2009). The AirHARP instrument was validated in the lab to perform at a 3 %–5 % radiometric and 0.5 % DoLP uncertainty across all spectral bands, though HARP2 may further reduce this uncertainty with improvements to onboard calibration and optical design (McBride et al., 2020). The study in this paper uses 3 % radiometric uncertainty for all the bands and 0.5 % DoLP uncertainty for 440, 550, and 670 nm and 1.5% for 870 nm as inputs to GRASP. The 870 nm polarimetric data have larger uncertainty due to a lower signal-to-noise ratio in the field data compared to 440, 550, and 670 nm; therefore, we give these data less relative weight in the retrievals. The total radiance ($I$) and DoLP are both useful for accuracy assessments and retrievals: they are not sensitive to a reference plane that defines electric field $E$. $Q$ and $U$ as measured by AirHARP, on the other hand, are defined based on a reference plane, and their absolute values depend directly on this chosen frame of reference. The details of this reference plane, including instrument scattering geometry, are described in the next section.

## 2.3 Definition of scattering geometry

Figure 2 defines the scattering geometry. Scattering angle ($\theta_{sca}$) is defined as the angle between the Sun vector and the viewing direction. $\theta_s$ is the solar zenith angle, $\theta_v$ is the instrument viewing zenith angle, $\phi_{sat}$ is the satellite azimuthal angle and $\phi_{sun}$ is the solar azimuthal angle. The relative azimuthal angle is $\phi = \phi_{sun} - \phi_{sat}$. For the calculation of $\theta_{sca}$, we need to know $\theta_s$, $\theta_v$ and $\phi$.

The reference plane for the definition of $E_\perp$ and $E_\parallel$ is based on the local meridian plane, which is a standard reference frame used for reporting $Q$ and $U$ (Chandrasekhar, 1950; Emde et al., 2015; Hansen and Travis, 1974; Hove-

**Table 2.** Table of flights analyzed from the ACEPOL campaign which flew over the ocean, land, forest fire smoke, and AERONET sites.

| Date of flight | Time of flights (UTC) | Target type |
|---|---|---|
| 23 October 2017 | 21:30–21:36 | Ocean |
| 23 October 2017 | 21:53–21:59 | Aeronet station (CalTech) |
| 25 October 2017 | 18:26–18:32 | Rosamond Dry Lake (California, USA) |
| 25 October 2017 | 19:55–20:01 | Aeronet station (Bakersfield) |
| 26 October 2017 | 18:55–20:01 | Aeronet station (Fresno_2) |
| 26 October 2017 | 19:15–19:21 | Aeronet station (Bakersfield) |
| 27 October 2017 | 18:16–18:22 | Smoke over land |
| 7 November 2017 | 18:09–18:15 | Aeronet station (CalTech) |
| 7 November 2017 | 19:36–19:42 | Aeronet station (Fresno_2) |
| 7 November 2017 | 20:03–20:09 | Aeronet station (Modesto) |
| 9 November 2017 | 18:31–18:37 | Aeronet station (USGS_Flagstaff_ROLO) |
| 9 November 2017 | 19:31–19:37 | Smoke over land |

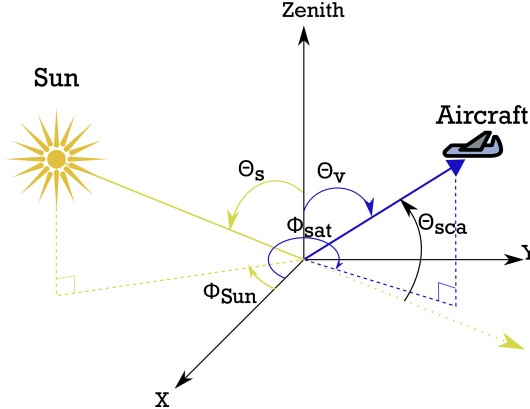

**Figure 2.** AirHARP viewing geometry and definition of angles. $\Theta_s$ is the solar zenith angle, $\Theta_v$ is the viewing zenith angle, $\Phi_{sun}$ is the solar azimuthal angle, $\Phi_{sat}$ is the viewing azimuthal angle, and $\Theta_{sca}$ is the scattering angle. The point where $X$ and $Y$ coordinates meet is the local ground point.

nier et al., 2004). For detailed information, please refer to the coordinate system as defined in the book by Hovenier et al. (2004). $Q$ and $U$ measured by AirHARP are based on the instrument reference frame, and these are rotated to the local meridian plane that is formed of the local nadir vector plus the viewing zenith vector (see Fig. 2). The electric field parallel to the local meridian plane is called $E_\parallel$ and the electric field perpendicular to the local meridian plane is $E_\perp$. Using the information from an aircraft's inertial measurement unit (IMU), the $Q$ and $U$ are rotated to the local meridian plane from the instrument reference frame.

## 2.4 Aerosol Characterization from Polarimeter and LiDAR (ACEPOL)

The ACEPOL campaign (https://www-air.larc.nasa.gov/missions/acepol/index.html, last access: 8 September 2020) was a collaborative effort of NASA and SRON (Netherlands Institute of Space Research) based out of Armstrong Flight Research Center (AFRC) in Palmdale, California, USA. One primary aim of the campaign was to acquire data using airborne advanced passive and active remote sensing instruments and then use the expanded information content available from the new sensors, both individually and in synergy, to better characterize aerosol (Knobelspiesse et al., 2020). Multiple polarimeters and lidars were mounted on the NASA ER-2 aircraft. These included AirHARP as well as AirMSPI (Diner et al., 2013), RSP (Cairns et al., 1999, 2003), SPEX airborne (Smit et al., 2019), HSRL2 (Burton et al., 2018; Hair et al., 2008) and Cloud Physics Lidar (CPL) (McGill et al., 2002). ACEPOL also made use of ground-based instruments such as AERONET and the Micro-Pulse Lidar Network (MPLNET) for validation of aircraft measurements (Holben et al., 2001; Welton et al., 2001). The measurements and inversion algorithms used to analyze the ACEPOL data will be helpful in understanding the potential use of polarimeters in future satellite missions like the NASA PACE mission, the Aerosols, Clouds, Convection and Precipitation (A-CCP) Decadal Survey mission, and the European EarthCare mission. The ACEPOL campaign had nine flights over the period of 19 October to 9 November 2017, with a combined flight time of approximately 41.3 h. The main objectives of ACEPOL include the calibration of instruments over ocean and land with no clouds or aerosol, geolocation of images using coastlines, coordinated Cloud-Aerosol Lidar and Infrared Pathfinder Satellite Observation (CALIPSO) or Cloud-Aerosol Transport System (CATS) under flights, validation with AERONET in low, medium and high aerosol loading, satellite intercomparison for aerosol and cloud retrievals, and calibration over a spatially uniform surface amongst other lower-priority goals, such as cirrus cloud observations (Knobelspiesse et al., 2020).

For the ACEPOL 2017 campaign, the AirHARP instrument was mounted on an ER-2 aircraft left-wing pod. It collected data over eight flights consisting of a total of 45 flight leg images. For this study, we have analyzed only 12 of these

flight legs, listed in Table 2, including scenes over the ocean, dry lake, forest fire smoke, and AERONET sites. Along with the airborne polarimeters, the HSRL2 also flew aboard the ER-2 aircraft during ACEPOL. HSRL2 is a NASA high spectral lidar that has been used to measure clouds and aerosols (Burton et al., 2012; Hair et al., 2008). HSRL2 measures extinction coefficients at two wavelengths ($2\alpha$ at 355 and 532 nm) and backscattering coefficients at three wavelengths ($3\beta$ at 355, 532, and 1064 nm) (Burton et al., 2018). These measurements allow the detection of the vertical distribution of aerosol extinction with a precision of about $0.02\,\mathrm{km}^{-1}$. The HSRL2 instrument points at nadir and measures the vertical profile of the aerosol backscatter coefficient at 0.1 Hz frequency with a vertical resolution of 15 m and an aerosol extinction coefficient at 1 min temporal resolution and 150 m vertical resolution. In some ACEPOL cases, due to atmospheric turbulence, interference in the HSRL2 measurement resulted in the inability to use the molecular channels at 355 and 532 nm to report the AOD and required assumed lidar ratios of 20 and 40 sr over the ocean and land, respectively. However, for all the comparisons shown in this study, those cases were avoided, and HSRL2 AOD reported here required no lidar ratio assumptions.

## 3 Aerosol retrievals from AirHARP using GRASP

### 3.1 Atmospheric gas absorption correction for aerosol retrieval

Before the inversion of the AirHARP-measured $I$, $Q$ and $U$ components, each measured pixel must be prepared for aerosol retrievals. This involves first avoiding groups of pixels that are inappropriate for an aerosol retrieval, such as clouds, and correcting for gaseous absorption in the remaining signal. Automatic algorithm-level cloud masking can be challenging. In the work presented here, scenes were selected by eye, so that there was no need to develop an automatic cloud mask for AirHARP at this time. However, correction for gaseous absorption was necessary. $I$, $Q$, and $U$ are corrected for the atmospheric gas absorption using the technique mentioned in Patadia et al. (2018). For AirHARP spectral bands, gas absorption is most significant at the 550 and 670 nm bands and is mainly due to the four atmospheric gasses $O_2$, $H_2O$, $O_3$, and $NO_2$. Columnar optical depths of 0.004, 0.032, 0.014, and 0.003 due to atmospheric gases are observed at the 440, 550, 670, and 870 nm spectral bands, respectively. The Unified Linearized Vector Radiative Transfer Model (UNL-VRTM) (Wang et al., 2014; Xu and Wang, 2019) is used to calculate transmission due to the total effect of all atmospheric gas absorption, which is translated to a multiplicative correction factor for each of the four AirHARP bands. These correction factors are a function of the path length through the atmosphere, which is a combination of solar and instrument viewing zenith. All the calculations are

done for a mid-latitude summer US atmosphere assuming no variation in the four gases and for an AirHARP observation height of 20 km a.s.l. This correction is applied to each band, for each pixel, prior to the inversion.

### 3.2 Generalized Retrieval of Aerosol and Surface Properties (GRASP)

GRASP is a versatile retrieval algorithm that can be used for a variety of remote sensing and in situ measurements to retrieve aerosol and surface properties (Dubovik et al., 2014). It is open-source software and is available free to non-commercial users for downloading from the website https://www.grasp-open.com/ (last access: 8 September 2020). GRASP first demonstrated its overall capability in an aerosol retrieval test study (Kokhanovsky et al., 2010). It has gone on to prove itself in a variety of real-world applications (Chen et al., 2018, 2019; Frouin et al., 2019; Li et al., 2019; Schuster et al., 2019; Torres et al., 2017). GRASP has been successfully applied to measurements from many different types of instruments (Benavent-Oltra et al., 2019; Espinosa et al., 2017, 2018; Román et al., 2018; Titos et al., 2019; Torres et al., 2017), but the most pertinent to AirHARP are the previous applications of GRASP to POLDER-3 on PARASOL (Chen et al., 2018, 2019; Dubovik et al., 2011; Frouin et al., 2019; Li et al., 2019) because of its familiar polarization, multi-wavelength, and multi-angle sampling characteristics.

GRASP consists of two modules: a forward model and an inversion module. For the case of aerosol measurements, the forward model consists of a polarized radiative transfer (RT) code to calculate the radiance measured by the instrument, and it uses a precalculated spheroidal kernel to calculate the contribution of single scattering by the aerosol particles following the strategy described by Dubovik et al. (2006, 2011). The kernel includes the pre-calculated full phase matrix elements, extinction and absorption for five log-normal size distributions with preselected size parameters for the ranges of the real refractive index 1.33 to 1.7 and 0.0005 to 0.5 for the imaginary part of the refractive index for both spherical and non-spherical aerosol approximated by a mixture of spheroids with a fixed particle shape distribution derived in Dubovik et al. (2006). This approach allows for very fast and accurate calculations of aerosol single-scattering properties in the wide range of refractive indices even for non-spherical aerosol. The details of the application of the kernels to satellite polarimetry are discussed in detail by Dubovik et al. (2011) (e.g., see Sect. 3.1 and Fig. 4 in Dubovik et al., 2011).

For the RT calculations, GRASP uses a successive order of scattering (SOS) scheme. The RT module consists of different surface bi-directional reflectance distribution function (BRDF) and bi-directional polarized distribution function (BPDF) models for land and ocean. These models are briefly discussed in Appendix A, and further information about the RT code and single-scattering database that is be-

**Table 3.** Five log-normal modes used for particle size distribution in GRASP retrieval for AirHARP; $r_v$ is the volume median radius and $\sigma_v$ the geometric standard deviation (SD). In this kernel, other particle properties are free to be retrieved.

| $r_v$ (μm) | $\sigma_v$ |
|---|---|
| 0.1 | 0.35 |
| 0.1732 | 0.35 |
| 0.3 | 0.35 |
| 1.0 | 0.5 |
| 2.9 | 0.5 |

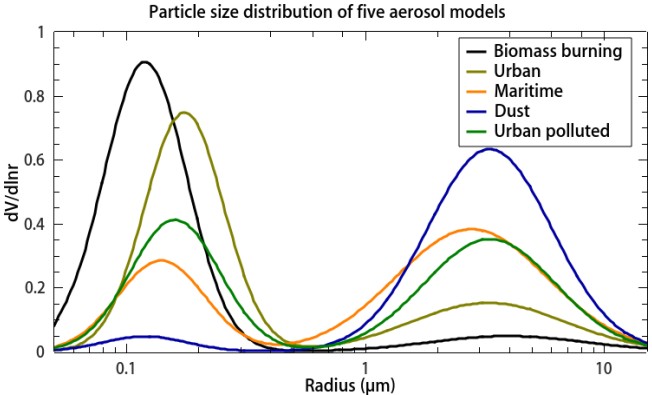

**Figure 3.** The particle size distribution of five aerosol components listed in Table 4. This simplified aerosol component-based GRASP/Models kernel is used for the AirHARP AOD inversion over the collocated AERONET pixels during the ACEPOL 2017 campaign.

yond the scope of this paper can be found in Dubovik et al. (2011) and Lenoble et al. (2007).

The particle single-scattering calculations that we employ for the AirHARP retrieval use one of two possible retrieval setups: (1) five log-normal distribution modes as described in Table 3 or (2) the aerosol assumed to be an external mixture of five aerosol components as described in Table 4. Both approaches were extensively used in PARASOL/GRASP processing and, therefore, are considered here. For the first kernel possibility, the retrieval has 15 aerosol parameters to retrieve and is called a "GRASP/Five mode" kernel. Each of the log-normal modes has a fixed mode radius and width. The free parameter in the retrieval for particle size distribution is the concentration of particles in each bin. There are three log-normal modes in the fine mode (log-normal modes 1 to 3 in Table 3) and two in the coarse mode (log-normal modes 4 and 5 in Table 3). Other retrieved parameters related to aerosol properties include a complex refractive index, aerosol layer height, and the fraction of spherical particles (SF). The same kernel is used for all the retrievals in this paper, with an exception for the AERONET comparison mentioned in Sect. 5.2. For the AERONET comparison, we make use of the second GRASP kernel that has reduced the number of aerosol parameters from 15 to 6. This reduced parameter option is the "GRASP/Models" kernel, where particle properties are assumed for each aerosol component given in Table 4. Complex refractive index, SF, and particle size distribution of each aerosol components are fixed for this kernel. Only the concentration (weight) for each aerosol component is retrieved.

The inversion module in GRASP uses the multi-term least square method (LSM) to solve the following system of equations:

$$f^* = f(a) + \Delta_f,$$
$$0^* = Ga + \Delta_g,$$
$$a^* = a + \Delta_a, \tag{3a}$$

TS5 where $a$ is a vector of unknowns and is called a *state vector*. $f^*$ is the vector which contains the instrument observations, $\Delta_f$ is the uncertainty in the observations, and $f(a)$ is the forward model simulated observations. For the AirHARP observations, $f^*$ is a vector containing information of $R_I$, $Q/I$ (same as $R_Q/R_I$), and $U/I$ (same as $R_U/R_I$) or $R_I$ and DoLP for all the spectral bands and viewing angles. GRASP is able to accept different configurations of the input parameters to make its retrieval. We will use the following sets of input in this work in different situations: $(R_I, Q/I, U/I)$ or $(R_I, \text{DoLP})$. The text will explicitly state the inputs in each instance. Given an ideal pixel, AirHARP measures 120 data points for each aforementioned variable. A priori smoothness constraints are imposed on the retrieved solution in order to suppress unrealistic oscillations in the retrieved characteristics. The second equation in Eq. (3a) represents such a smoothness constraint on the retrieved characteristics, and $0^*$ is the zero vector and imposes the forced constraint that the derivatives of retrieved parameters be zero. The matrix $G$ includes the coefficients for calculating derivatives of state vectors approximated by finite differences. For example, unrealistic oscillations in particle size distribution are eliminated using coefficients calculated from derivatives with respect to radius. Similarly, spectral dependencies of the refractive index are imposed using the coefficients calculated using wavelength. Uncertainties in the smoothness constraints are represented by the $\Delta_g$ term. GRASP can perform retrievals using multi-pixel information in both spatial and temporal dimensions; however, in this study, we are not utilizing this feature due to the limited availability of data over the same place in the temporal dimension. We use a priori constraints on the particle size distribution, and these constraints are represented by the third and last term in Eq. (3b). A priori estimates of state vector parameters are given by $a^*$ and $\Delta_a$ is the uncertainty in the a priori constraints of $a^*$. The multi-term LSM in GRASP finds the statistically optimized solution of the set of equations mentioned in Eq. (3a) by minimizing the

**Table 4.** Details on the complex refractive index and particle size distribution parameters used for the aerosol models. $C_{\mathrm{v}}$ is the concentration, $\sigma_{\mathrm{v}}$ is the SD of the distribution, and $r_{\mathrm{v}}$ is the volume median radius. Fine- and coarse-mode parameters are indicated by $f$ and $c$ subscripts, respectively.

| Aerosol models | Real part of the refractive index (RRI) | Imaginary part of the refractive index (IRI) (440, 550, 670, 870 nm) | | | | $C_{\mathrm{v,f}}$ | $\sigma_{\mathrm{v,f}}$ | $r_{\mathrm{v,f}}$ μm | $C_{\mathrm{v,c}}$ | $\sigma_{\mathrm{v,c}}$ | $r_{\mathrm{v,c}}$ μm |
|---|---|---|---|---|---|---|---|---|---|---|---|
| Biomass | 1.510 | 0.050 | | | | 0.91 | 0.40 | 0.120 | 0.09 | 0.75 | 3.95 |
| Urban | 1.395 | 0.003 | | | | 0.75 | 0.38 | 0.018 CE1 | 0.30 | 0.75 | 3.27 |
| Urban polluted | 1.470 | 0.100 | | | | 0.52 CE2 | 0.43 | 0.160 | 0.48 CE3 | 0.63 | 3.32 |
| Maritime | 1.370 | 0.0001 | | | | 0.30 | 0.42 | 0.140 | 0.70 | 0.73 | 2.78 |
| Dust | 1.560 | 0.0029 | 0.0019 | 0.0013 | 0.0011 | 0.02 | 0.40 | 0.120 | 0.40 CE4 | 0.60 | 3.32 |

term: TS6

$$2\Psi(\boldsymbol{a}) = \left(f(\boldsymbol{a}) - \boldsymbol{f^*}\right)^{\mathrm{T}}\mathbf{W_f}^{-1}(f(\boldsymbol{a}) - \boldsymbol{f^*})$$
$$+ \gamma_{\mathrm{g}}\boldsymbol{a}^{\mathrm{T}}\mathbf{W_g}^{-1}\boldsymbol{a} + \gamma_{\mathrm{a}}(\boldsymbol{a} - \boldsymbol{a^*})^{\mathrm{T}}\mathbf{W_a}^{-1}(\boldsymbol{a} - \boldsymbol{a^*}), \qquad (3b)$$

where $\mathbf{W_f} = \frac{1}{\epsilon_f^2}\mathbf{C_f}$ is the weighting matrix calculated using the measurement covariance matrix $\mathbf{C_f}$ and the first diagonal element $\epsilon_f$ in $\mathbf{C_f}$. Similarly, $\mathbf{W_g} = \frac{1}{\epsilon_g^2}\mathbf{C_g}$, $\mathbf{W_a} = \frac{1}{\epsilon_a^2}\mathbf{C_a}$ are calculated using the covariance matrices of a priori smoothness constraints and a priori constraints on the retrieved parameters. $\gamma_{\mathrm{g}} = \frac{\epsilon_f^2}{\epsilon_g^2}$ and $\gamma_{\mathrm{a}} = \frac{\epsilon_f^2}{\epsilon_a^2}$ are the *Lagrange multipliers* (Phillips, 1962; Tikhonov, 1963) calculated using the information from the covariance matrices. In order to take into account the non-negative character of measured and retrieved physical values in the retrieval optimization, the log-normal error distributions are assumed for all positively defined measured characteristics, and the minimization is defined for logarithms of all positively defined retrieved parameters. The solutions to the set of equations in Eq. (3a) are found by minimizing the term $\Psi(\boldsymbol{a})$ in Eq. (3b). Since the radiative transfer in the atmosphere has a pronounced nonlinear character, the Levenberg–Marquardt (Levenberg, 1944; Marquardt, 1963) algorithm is harmoniously adapted into the statistically optimized fitting to ensure the monotonic convergence of the iterative solution. These and other technical details of numerical inversion are described in Dubovik et al. (2011), and in-depth discussion of the above methodological aspects also can be found in Dubovik and King (2000) and Dubovik (2006).

The state vector $\boldsymbol{a}$ includes the information on particle size distribution, which is the concentration for five log-normal modes of Table 3, the complex refractive index in the four spectral bands that are independent of particle size, the SF, aerosol layer height, and parameters characterizing the directional reflectance of the surface. AOD is derived from retrieved aerosol properties using the method mentioned in Appendix B1. Additionally, fine-mode AOD is calculated using modes 1–3 mentioned in Table 3 and coarse-mode AOD using modes 4 and 5. Single scattering albedo (SSA) and Angstrom exponent (AE) are also derived from the retrieved aerosol properties. For the GRASP/Models approach, the state vector $\boldsymbol{a}$ includes the concentration for each aerosol component mentioned in Table 4. State vector $\boldsymbol{a}$ does not contain information on the particle size distribution, SF, and complex refractive index. All this is embedded in the aerosol components which are close (with some modifications) to biomass burning, urban, urban polluted, maritime, and desert dust observed in AERONET climatology by Dubovik et al. (2002). Among these, only desert dust is considered completely non-spherical and, similarly to AERONET retrievals, uses a shape distribution mentioned in Dubovik et al. (2006). All the other types are treated as 100 % spherical particles. The details of the bi-modal size distribution parameters along with the fixed complex refractive index for each of the aerosol components are tabulated in Table 4 and are based on the work of Dubovik et al. (2002). Figure 3 shows the particle size distribution as a function of radii for the different aerosol components. The main differences between the GRASP/Five mode and GRASP/Models approaches are that (1) instead of retrieving the concentration of each log-normal mode, concentration (weight) for each of the aerosol components mentioned in Table 4 is retrieved. (2) RRI, IRI, and SF are not retrieved since these are fixed for each of the aerosol components. This simplified approach significantly drops the complexity of the aerosol model by reducing the number of parameters retrieved directly in the joint retrieval. It helps in reducing the nonlinearity of the inverse problem and makes the separation of the surface and aerosol signal much less complicated compared to the GRASP/Five mode approach. At the same time, all aerosol total properties, such as SSA, effective size distribution, and complex refractive index, can be obtained using an external aerosol mixture concept. The reduction of sought unknowns helps situations in lower information content (e.g., for low AOD) and makes the separation of the surface and aerosol signal much less complicated compared to a GRASP/Five mode kernel. This tendency is well identified in the in-depth analysis of PARASOL data processing using different retrieval setups by Chen et al. (2020). Like the GRASP/Five mode kernel, the state vector $\boldsymbol{a}$ includes the information on aerosol layer height. Even though aerosol layer

height is retrieved during the retrieval process, the sensitivity to aerosol height for the AirHARP wavelengths is negligible for most of the low loading cases. Retrieved aerosol layer height is thus not discussed in this work.

Retrieval of aerosol properties from MAPs is highly sensitive to the accurate representation of the directional reflectance from the surface. For the ocean pixels, the surface model used is the NASA GISS model (Mishchenko and Travis, 1997) based on Cox and Munk (1954), in which the ocean surface reflectance is represented by three parameters: ocean surface albedo, the fraction of the Fresnel reflection surface, and wind speed denoted by $a_0$, $a_1$, and $a_2$, respectively, with details given in Appendix A2. These three parameters are the surface components in the state vector $\boldsymbol{a}$ for the case of ocean pixels. For the case of land pixels, the Ross-thick Li-sparse linear BRDF model is used to represent the directional reflectance from the surface (see Appendix A1.1), which uses three parameters $K_0$, $K_1$, $K_2$. $K_0$ is a spectrally dependent parameter that represents the isotropic reflectance, and $K_1$ and $K_2$ normalized to $K_0$ are the spectrally independent parameters which are the coefficients of geometric and volumetric scattering kernels, respectively (Maignan et al., 2004; Wanner et al., 1995). Polarized reflectance from the surface is modeled using the Maignan–Breon one-parameter model, and the retrieved parameter is a scaling factor ($\alpha$) that is spectrally dependent (Maignan et al., 2009). Refer to Appendix A1.2 for detailed information on the surface models. The four parameters $K_0$, $K_1/K_0$, $K_2/K_0$, and $\alpha$ are the surface components in the state vector $\boldsymbol{a}$ for the case of land pixels. In the next section, we discuss the results of applying GRASP to AirHARP measurements of $R_I$, $Q/I$, and $U/I$ for selected cases from the ACEPOL campaign.

## 4 Aerosol properties from AirHARP measurements

### 4.1 Selected cases from ACEPOL 2017

We will focus on four specific cases from ACEPOL 2017 to illustrate the measurement characteristics of AirHARP and demonstrate GRASP retrieval. Figure 4 shows color composite imagery of the intensity and DoLP of each case, using the 440, 550, and 670 nm bands. Cases include a scene over the ocean with little aerosol loading (23 October 2017; T21:30 UTC), a scene over Rosamond Dry Lake (25 October 2017; T18:26 UTC), and two scenes of forest fire smoke (27 October 2017; T18:16 UTC, and 9 November 2017; T19:30 UTC).

### 4.2 Scene over the ocean (23 October 2017 T21:30 UTC)

The first of our analysis scenes is a cloud-free segment where the ER-2 flew over the USC SeaPRISM AERONET station located at 33.564° N, 118.118° W, on a platform off the coast of southern California. The AERONET station measured a low aerosol loading of AOD = 0.04 at 440 nm at the time of ER-2 overpass. The segment includes sunglint, and because of the low AOD, the sunglint and non-sunglint patterns are ideal for the intercomparison of different polarimeter measurements of $I$, $Q$, $U$, and DoLP. This flight track aligned with the solar principal plane so that the longer wavelength bands will be highly polarized for the sunglint viewing angles. For a scene with low aerosol loading above the ocean with no sunglint, the polarization follows the Rayleigh pattern and will peak at the 90° scattering angle. For the sunglint case, we expect the peak to be at scattering angles where the sunglint is observed. In this case, the maximum sunglint occurs for scattering angles 70 to 90°. Figure 5 shows the measured $R_I$, $Q/I$, $U/I$, and DoLP for 1 pixel (footprint of 55 m × 55 m) along the nadir track as a function of scattering and plotted as colored circles, for each of the four wavelengths. The measurements show that the maximum intensity occurs in the glint region (scattering angles 70 to 90°) and confirms that the DoLP peak also occurs at the sunglint scattering angles. However, while the intensity falls to minimum levels outside of the glint region, the DoLP has a more gradual falloff, as the Rayleigh pattern with maximum DoLP at 90° is superimposed on the dark ocean scene.

GRASP is applied to invert the measured $R_I$, $Q/I$ and $U/I$ for the pixel represented in Fig. 5 for aerosol retrievals. Because the aerosol loading is very low for this scene, there is insufficient aerosol loading to retrieve the real (RRI) and imaginary (IRI) parts of the complex refractive index, and instead they are constrained (RRI = 1.4 and IRI = 0.0001) in GRASP using the values of the oceanic aerosol model mentioned in Hasekamp et al. (2008). This will reduce the number of retrieved parameters and thus reduce the complexity of the inversion problem by reducing the nonlinearity of the forward model. The GRASP/Five mode kernel is used in the retrieval, with the concentrations of each of the five modes unconstrained. Therefore, the retrieved parameters include the five concentrations for the five log-normal modes shown in Table 3, aerosol spherical fraction, aerosol layer height, and the ocean model parameters $a_0$, $a_1$, and $a_2$. AOD is derived from the retrieved and modeled parameters. The solid black lines plotted in all panels of Fig. 5 are the GRASP fits using the AirHARP-measured $R_I$, $Q/I$, and $U/I$ as input. The DoLP is also calculated from the fitted variables and plotted in the same figure. The sunglint registers in $R_I$ as a sharp peak, with the width of that peak dependent on surface roughness primarily caused by surface wind. The retrieval of aerosol properties is highly sensitive to wind speed. An inappropriate wind speed estimate can result in high uncertainty in the aerosol properties retrieved. The goodness of fit in Fig. 5 suggests that retrieval of the ocean parameters, including wind speed, is very good for this sampled pixel.

To achieve a better understanding of how well the GRASP retrieval can fit the measurements, we apply the retrieval to 3600 pixels (60 × 60) of this ocean scene. Here, a pixel footprint of 55 m × 55 m is used, since the variability due to the

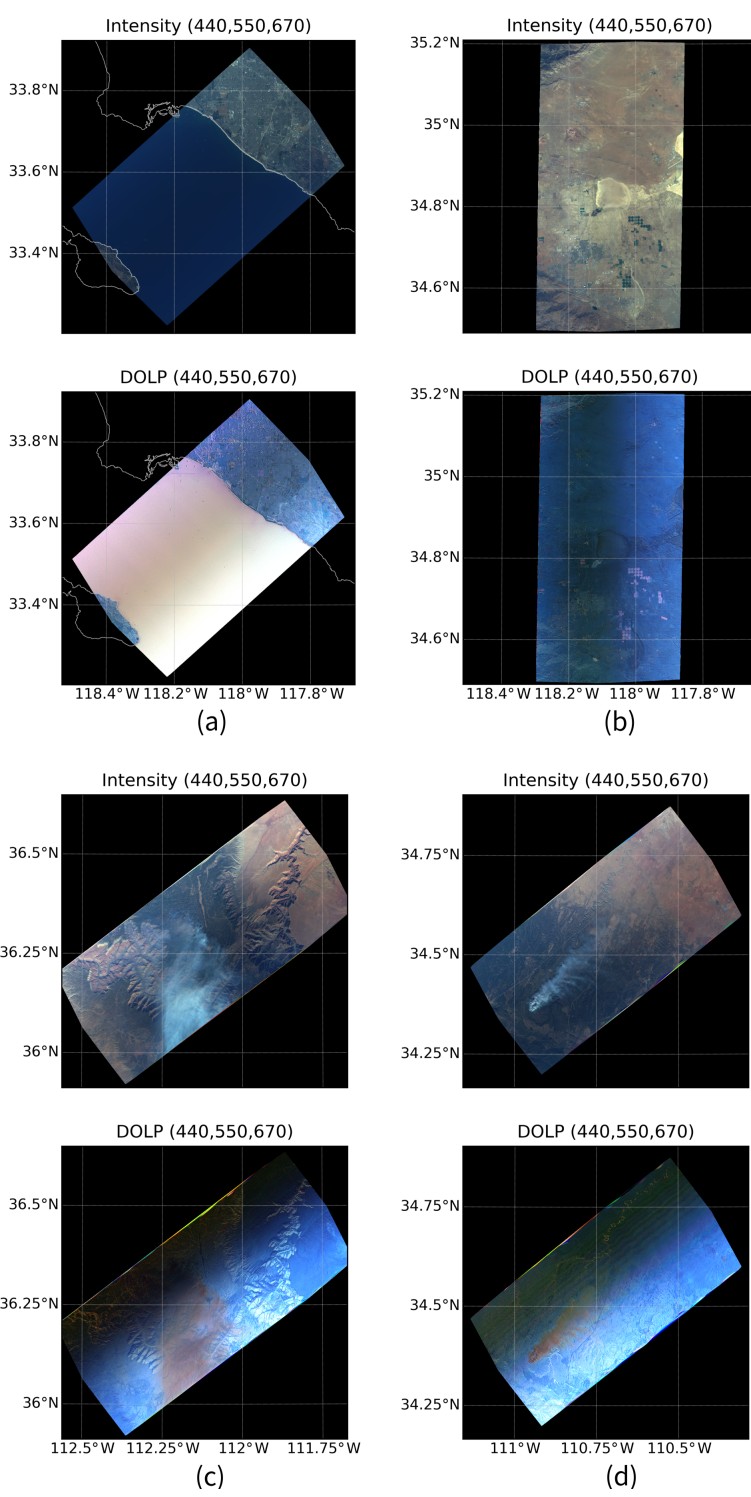

**Figure 4.** Nadir RGB (670, 550, and 440 nm) images of radiance and DoLP for the selected cases from AirHARP flight legs listed in Table 2. The $X$ axis is the longitude and the $Y$ axis the latitude. The scenes include **(a)** sunglint scene over the Pacific Ocean near Los Angeles, California, USA (23 October 2017 T21:30 UTC); **(b)** Rosamond Dry Lake in California, USA (25 October 2017 T18:26 UTC); **(c)** biomass burning scene near the Kaibab National Forest and Grand Canyon National Park (27 October 2017 T18:16 UTC); and **(d)** biomass burning scene near the Kaibab National Forest (9 November 2017 T19:30 UTC).

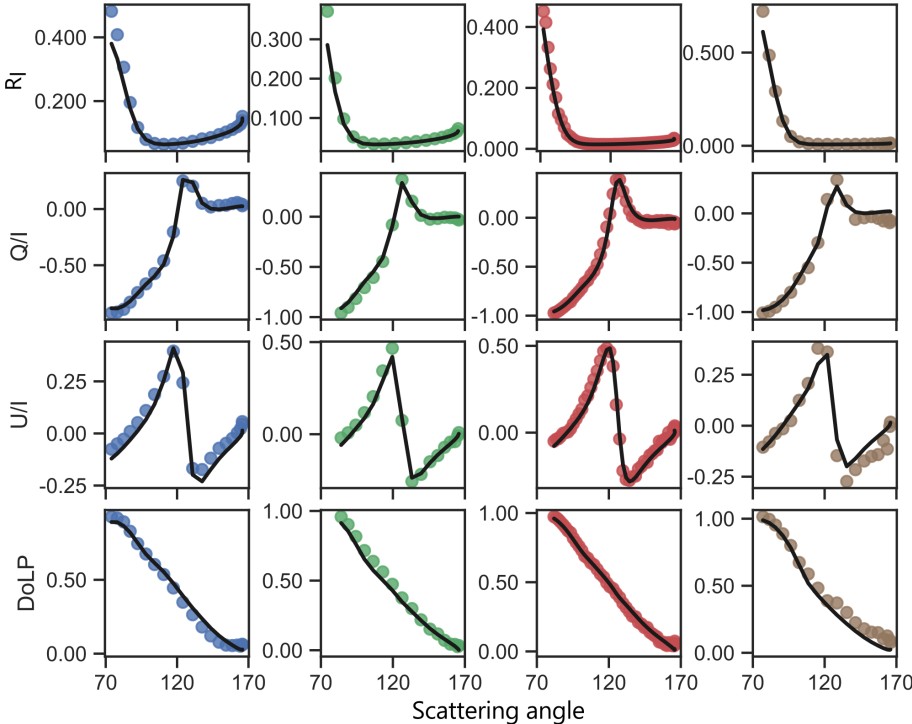

**Figure 5.** AirHARP-measured $R_I$, $Q/I$, $U/I$, and DoLP (solid colored circles) for a single sunglint pixel from the scene in Fig. 4a and the GRASP fit to the measurements (black solid line) for all bands (blue – 440 nm, green – 550 nm, red – 670 nm and brown – 870 nm).

geolocation is negligible for an open-ocean pixel as compared to a land pixel. Figure 6 shows 2-D density scatter plots of the AirHARP-measured variables $R_I$, $Q/I$, and $U/I$ for the four spectral bands vs. the GRASP fit, and the histogram of the number of points used for each bin is plotted on the respective axes. A dashed magenta line represents the ordinary least square (OLS) fit between retrieved and measured parameters; a black solid line denotes the 1 : 1 line. The goodness of fit, $\chi^2_{\mathrm{norm}}$ for each pixel, is calculated using the mathematical expression mentioned in Table 5. A list of the statistical parameters used in this study is formulated in Table 5, where $x_i$ is the measured value and $y_i$ is the GRASP fit. $N$ is the total number of observations for the pixel; $\mathbf{S_y}$ is the error covariance matrix for the observations. $\mathbf{S_y}$ includes only diagonal elements, and off-diagonal elements are assumed to be zero since we do not consider the cross-correlation between the different viewing angles for the same spectral band. For the reduced radiance $R_I$, all spectral bands show a good comparison at lower $R_I$ values, within the $\chi^2_{\mathrm{norm}}$ confidence interval, and show a slight deviation from the 1 : 1 line for the sunglint angle data points. The GRASP fit in blue band yields an underestimated $R_I$ when its values are greater than 0.2, giving an overall OLS slope of 0.967. However, the range where the underestimation occurs represents only a small fraction of the total analyzed samples. This underestimation is also observed for the 550 nm band, with a slightly better $R^2$ value of 0.995 compared to 0.986 for

the 440 nm band. For the case of the 670 and 870 nm bands, we observed a slight overestimation with the GRASP fitting having $R^2$ values 0.997 and 0.996 for red and near-infrared (NIR) bands, respectively. Some of this underestimation and overestimation is because the isotropic wind model has trouble simulating the multi-angle views in the sunglint region. In terms of the spread of points around the regression line, we expected much higher noise for the 870 nm band due to the lower signal-to-noise ratio (SNR) in this band. Surprisingly, in terms of fitting the $R_I$ component, the 870 nm band does not display any repercussion of the lower SNR. For the polarization components $Q/I$ and $U/I$, all the spectral bands demonstrate a good correlation between the GRASP fit and AirHARP measurements. This demonstrates that the polarization variables are less affected by the discrepancies in sunglint pixels for the extreme viewing angles. The average AODs retrieved for these 60 pixel by 60 pixel regions are $0.07 \pm 0.03$, $0.04 \pm 0.02$, $0.03 \pm 0.01$, and $0.02 \pm 0.01$ at 440, 550, 670, and 870 nm, respectively. In the following section, we detail several case studies of AirHARP land surface and aerosol plume data applied to GRASP for retrieval of aerosol microphysical and optical properties.

## 4.3 GRASP retrieval over land

Equally important over the land for a multi-angle instrument is the need to co-register each along-track view angle of the same target. Over the flat ocean, co-registration is straight-

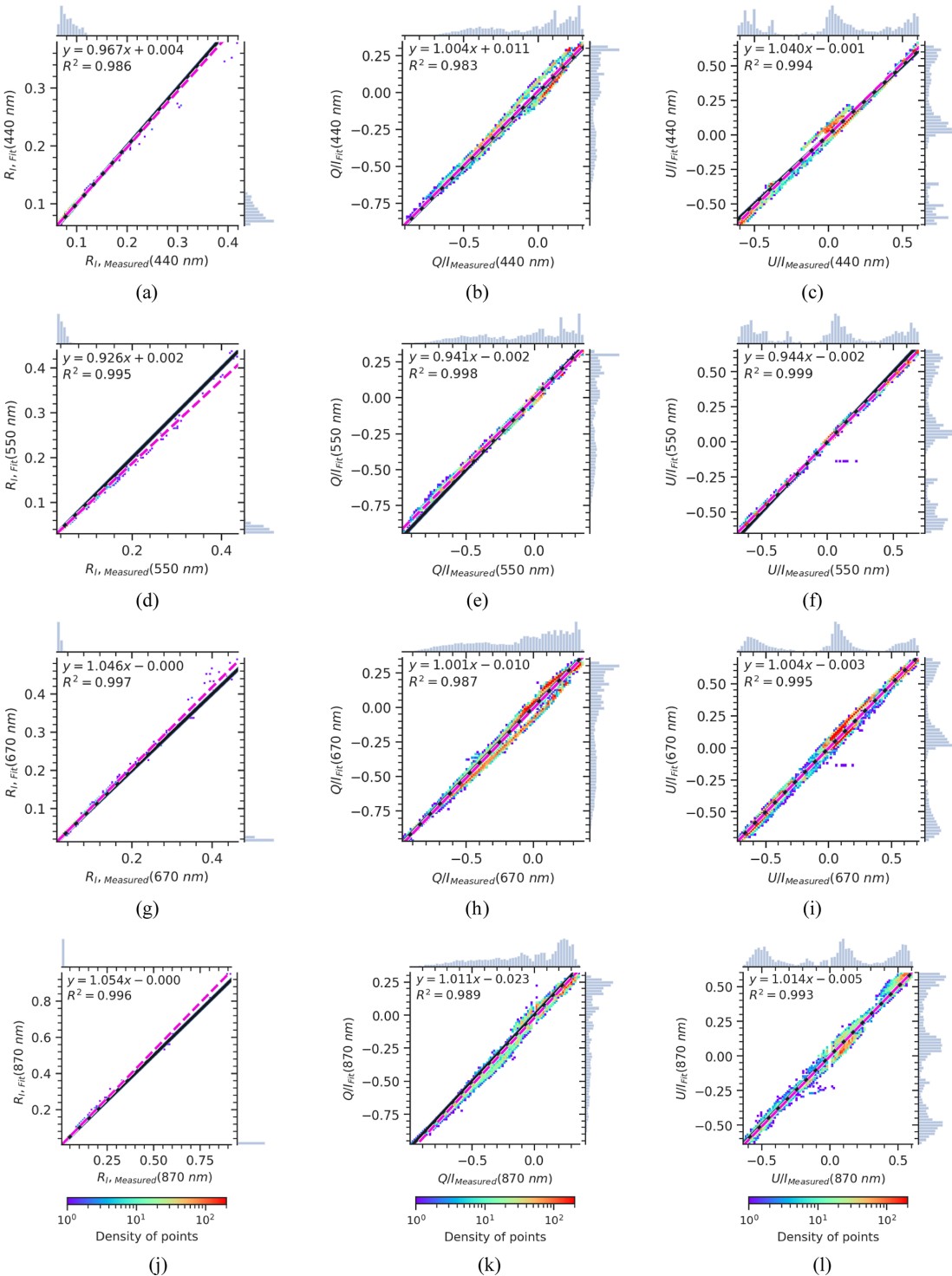

**Figure 6.** Scatter density plots for AirHARP measurement variables and GRASP fit for the ocean scene in Fig. 4a. The subplots **(a, d, g, j)** are the 2-D density plots for variable $R_I$ at different spectral bands blue, green, red and NIR, respectively; **(b, e, h, k)** are for the variable $Q/I$; **(c, f, i, l)** are for the variable $U/I$. The dashed magenta line is the ordinary least square (OLS) fit for the measured and GRASP-fitted variable. The solid black line is the 1 : 1 line for the measured and fitted variable. There are 72 000 data points for the 440, 550, and 870 nm bands and 216 000 data points for the 670 nm band. For each plot, the histogram of measurement and GRASP fit for each variable are plotted on the top and right axes, respectively.

**Table 5.** Definition of statistical parameters used in this study to find the correlation between measurement and models.

| Statistical parameter | Mathematical formulae |
|---|---|
| The goodness of fit ($\chi^2_{\text{norm}}$) | $\chi^2_{\text{norm}} = \frac{1}{N} \sum_{i=1}^{N} \frac{(y_i - x_i)^2}{S_y}$ |
| Pearson coefficient ($\rho$) | $\rho = \frac{\sum_{i=1}^{n}(x_i - \overline{x})(y_i - \overline{y})}{\sqrt{\sum_{i=1}^{n}(x_i - \overline{x})^2 \sum_{i=1}^{n}(y_i - \overline{y})^2}}$ |
| Coefficient of determination ($R^2$) | $R^2 = \rho^2$ |
| Mean absolute error (MAE) | $\text{MAE} = \frac{\sum_{i=1}^{N}(|y_i - x_i|)}{N}$ |
| Bias | $\text{BIAS} = \overline{y} - \overline{x}$ |

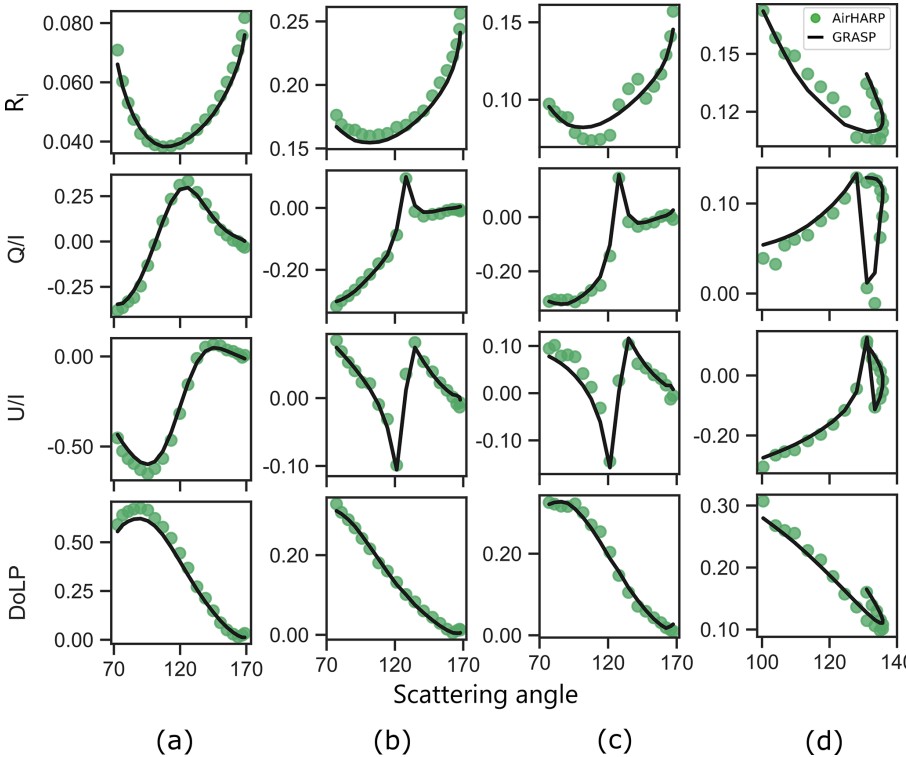

**Figure 7.** AirHARP-measured $R_I$, $Q/I$, $U/I$, and DoLP (solid colored circles) and the GRASP fit (black solid lines) for selected pixels from the scene in Fig. 4a–c are plotted for the 550 nm band. Column **(a)** is an offglint pixel from the scene on 23 October 2017 T21:30 UTC; **(b)** for a dry lake pixel from the flight leg on 25 October 2017 T18:26 UTC; **(c)** for a vegetation surface also from the flight leg on 25 October 2017 T18:26 UTC; **(d)** for a smoke pixel from the flight leg on 27 October 2017 T18:16 UTC.

forward and is based on a projection of the measurements onto a representation of a smooth geoid Earth. Over the land, topography introduces a challenging situation in which forward and aft views of the same target might image different slopes of a ridge. Topographically corrected projections need to be made either to a digital elevation model at a resolution comparable to the measurements (this is operational for AirHARP Level 1B data) or the measurements need to be projected to a specific altitude in the atmosphere, perhaps cloud top height or an aerosol layer. Figure 7 shows the mea-

sured $R_I$, $Q/I$, $U/I$, and DoLP from the 550 nm wavelength for selected pixels in each of the following three flight legs under analysis. In this figure, unlike Fig. 5, the ocean scene is from an offglint pixel. The other pixels represent a dry lake surface, vegetation, and smoke, respectively. Also plotted as the black curves are the GRASP fit to each of these targets. The ocean pixel appears the easiest to fit, and then the smooth dry lake pixels. The other land surface types, with their variable topography, present a greater challenge for GRASP. In the next section, we detail these three flight seg-

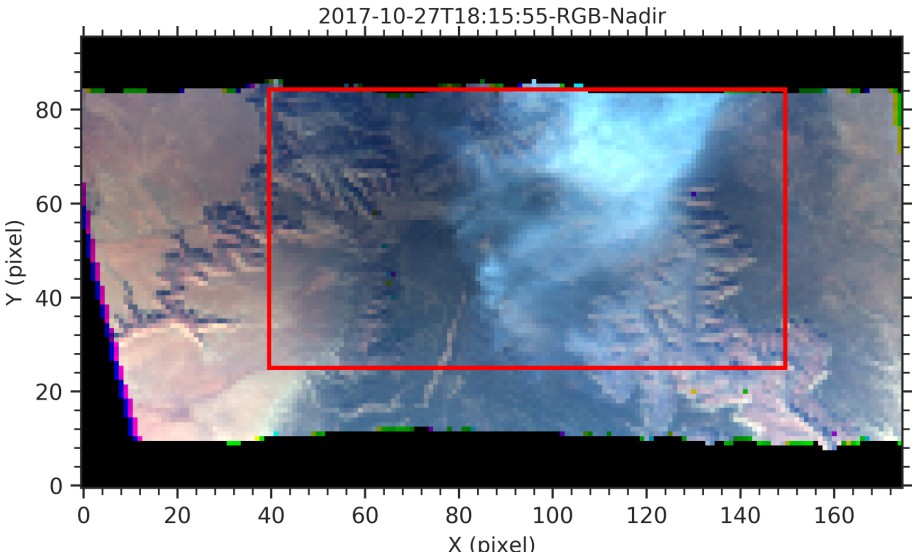

**Figure 8.** RGB composite image of the 27 October 2017 T18:16 UTC smoke scene; $X$ and $Y$ axes are the number of pixels (550 m × 550 m) along the respective axes. Pixels inside the red rectangular box are used for the aerosol retrievals of AOD, SSA, RRI, and IRI. These are plotted in Fig. 9.

ments over land that include one on 25 October 2017 over Rosamond Dry Lake at 18:26 UTC, a second one that is a forest fire smoke scene near the Kaibab National Forest and Grand Canyon National Park in Arizona, USA, on 27 October 2017 at 18:16 UTC, and a third scene of fresh smoke on 9 November 2017 at 19:30 UTC.

### 4.3.1  Rosamond Dry Lake and surrounding vegetation (25 October 2017 T18:26 UTC)

The flight leg with Rosamond Dry Lake on 25 October 2017 T18:26 UTC is a scene with very low aerosol loading according to the Moderate Resolution Imaging Spectroradiometer (MODIS) retrieval of AOD at 550 nm = 0.03. The minimal AOD allows for the measured signal to be dominated by surface reflectance features. The retrieved AODs of the pixel of the dry lake whose measurements and GRASP fit are shown in Fig. 7b were 0.05, 0.04, 0.04, and 0.04 for the 440, 550, 670, and 870 nm bands, respectively. Measurement and GRASP fit for another pixel from the same scene but over nearby vegetation (farm field) are plotted in Fig. 7c, and the retrieved AODs are 0.04, 0.03, 0.03, and 0.02 for the 440, 550, and 870 nm bands, respectively. Note that GRASP retrieves a very similar atmosphere over both surfaces. However, in Fig. 7b the homogenous surface of the dry lake provides a simpler retrieval, and the result is a better fit to the measurements than the more complex surface presented by the vegetation. The RGB image of the flight leg on 25 October 2017 at 18:26 UTC is plotted in Fig. 4b.

### 4.3.2  Forest fire smoke (27 October 2017 T18:16 UTC)

Up to now, all our examples have demonstrated AirHARP measurements and GRASP retrievals in very low aerosol loading. These situations can demonstrate GRASP's ability to fit the measurements and to return values for spectral AOD. The low aerosol loading does not supply enough signal to fully characterize the aerosol using GRASP. The final example that we show, the flight leg on 27 October 2017 at 18:16 UTC, captures a fire and smoke plume with significant aerosol loading. This will demonstrate the potency of AirHARP/GRASP to characterize aerosol particle properties, along with loading. This case has complicated terrain and, due to the higher resolution of the pixels, the aerosol retrievals from this scene will be complicated.

As a quick check, we show the GRASP retrieval fit to the input measurements in Fig. 7d for a single pixel in the smoke plume where the terrain is not that complicated and homogenous smoke is observed. The retrieval fits to the measurements well. AODs retrieved at this pixel are 1.62, 1.2, 0.85, and 0.51 for the 440, 550, 670, and 870 nm bands, respectively.

An RGB image of the smoke scene on 27 October 2017 at 18:16 UTC is plotted in Fig. 8 with the $X$ and $Y$ axes as pixel locations. The GRASP algorithm is applied to 7150 pixels in a large area marked by the red rectangle as shown in Fig. 8, and retrievals for the whole segment are plotted in Fig. 9. Measured $R_I$ and DoLP are used for fitting in the GRASP algorithm since these two variables are not sensitive to the definition of the local meridian plane, whereas $Q/I$ and $U/I$ are sensitive to the plane of reference for polarization, and this can introduce retrieval error due to the error in the rotation

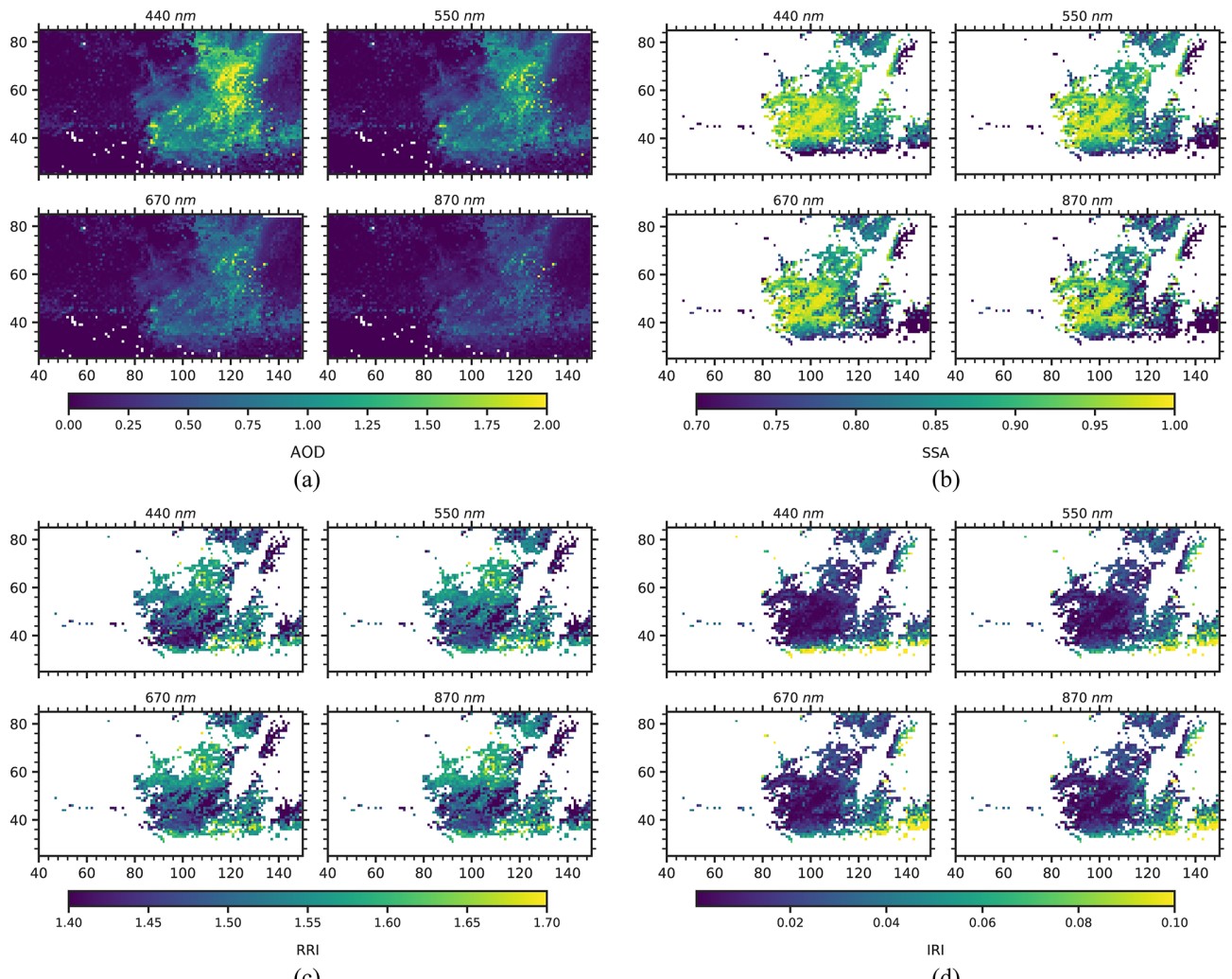

**Figure 9. (a)** AOD map of a subsetted portion of the 27 October 2017 T18:16 UTC smoke scene that was marked by a red rectangular box in Fig. 8; **(b)** single scattering albedo (SSA) for the same subsetted portion in **(a)**, but the pixels with the goodness of fit, $\chi^2_{\mathrm{norm}} > 5$ and AOD$_{440\,\mathrm{nm}} < 0.4$ are masked. **(c)** Same as **(b)** except the real part of the refractive index (RRI) is plotted. **(d)** Same as **(b)** except the imaginary part of the refractive index (IRI) is plotted. $X$ and $Y$ axes are the pixel coordinates.

of $Q$ and $U$ from the instrument reference plane to the local meridian plane. To avoid an extra source of uncertainty, $R_I$ and DoLP are used as the input in GRASP for large-scale retrievals. One exception is that for the ocean pixel study using the flight leg on 23 October 2017 T21:30 UTC we have used $R_I$, $Q/I$, and $U/I$ for the retrieval. This leg has been thoroughly quality checked for the error in the rotation of $Q$ and $U$ from the instrument reference plane to the local meridian plane. Retrievals include AOD, RRI, IRI, and SSA at the four spectral bands of AirHARP. AOD is plotted across the image, but the intrinsic particle properties are only shown where GRASP recognizes enough aerosol loading to be sensitive to particle properties. Thus, the plots follow the smoke plume. For the retrievals, a combination of Ross–Li and Maignan–Breon land surface BRDF and BPDF models is used to rep-

resent the directional reflectance from the land surface. The pixels are spatially averaged to a resolution of $550\,\mathrm{m} \times 550\,\mathrm{m}$ to avoid the micro-pixel movement effects which will affect the aerosol retrievals. High non-homogenous smoke near the source makes accurate aerosol retrieval difficult. This is because the GRASP assumes a plane-parallel aerosol layer in the radiative transfer multi-angle calculations, whereas in reality at different viewing angles we are seeing different locations in the plume, which introduces complications into the radiative transfer calculations.

Figure 9a shows the retrieved AODs for the 440, 550, 670, and 870 nm spectral bands. The AOD at the 440 nm band is much higher than the one at the 870 nm band, as we expect for the fine forest fire particles. For the higher confidence in the retrieved results, pixels with $\chi^2_{\mathrm{norm}} > 5$ and AOD$_{440\,\mathrm{nm}} <$

**Table 6.** Mean aerosol optical and microphysical properties retrieved for the smoke scene in Fig. 9 (for pixels with $AOD_{440\,nm} > 0.4$).

| Spectral band | Single scattering albedo | Spherical fraction (%) | Angstrom exponent[a] | Real refractive index (RRI) | Imaginary refractive index (RRI) |
|---|---|---|---|---|---|
| 440 nm | $0.87 \pm 0.06$ | | | | |
| 550 nm | $0.86 \pm 0.07$ | $49.9 \pm 36\,\%$[b] | $1.53 \pm 0.336$ | $1.55 \pm 0.04$ | $0.024 \pm 0.017$ |
| 670 nm | $0.84 \pm 0.08$ | | | | |
| 870 nm | $0.81 \pm 0.09$ | | | | |

[a] Angstrom exponent calculated using the AOD at wavelength bands 440 and 870 nm of the AirHARP.
[b] Retrieved spherical fraction includes a significant number of pixels with SF $\sim 99\,\%$.

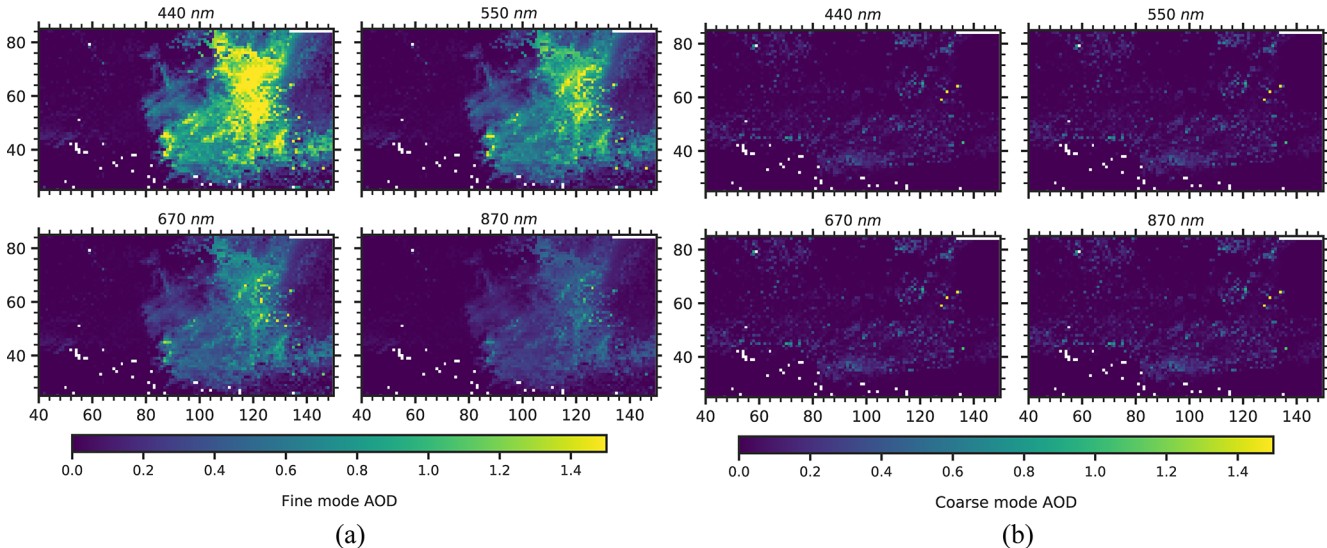

**Figure 10. (a)** Fine-mode AOD map of a subsetted portion of the 27 October 2017 T18:16 UTC smoke scene that was marked by a red rectangular box in Fig. 8 for all AirHARP spectral bands; **(b)** same as **(a)** but for coarse-mode AOD. $X$ and $Y$ axes are the pixel coordinates.

0.4 are removed from the analysis of SSA, RRI, IRI, AE, aerosol volume concentration, and SF. Retrieved forest fire smoke optical properties from the flight leg on 27 October 2017 at 18:16 UTC using AirHARP observations are close to values seen previously in the literature. The values of RRI retrieved from AirHARP and shown in Fig. 9c can be represented as a Gaussian distribution with a mode value of 1.55 for all wavelengths, while retrievals from the RSP and SPEX airborne instruments during the ACEPOL campaign produced values of RRI of 1.56 and 1.58, respectively, for a similar forest fire smoke (Fu et al., 2020). Fire Laboratory at Missoula Experiments (FLAME) records the real part of the refractive index in a range from 1.55 to 1.8, depending on the composition of the smoke particles (Poudel et al., 2017). FLAME 2 laboratory experiments reported RRI values in the range of 1.54 to 1.67, depending on the fuel source (Mack et al., 2010). The AirHARP retrievals of IRI range from 0.01 to 0.1 with a mean value of 0.024 (Fig. 9d), which compares well with the FLAME database range of 0.01 to 0.5 and the FLAME 2 database range of 0.011 to 0.217 (Mack et al., 2010). Another smoke optical property study reports SSA in

the range of 0.78 to 0.94 at 532 nm, depending on the age of the smoke (Nicolae et al., 2013). Our retrieved SSA from AirHARP in this work (see Fig. 9b) ranges from 0.6 to 0.99 with mean values of $0.87 \pm 0.06$, $0.86 \pm 0.07$, $0.84 \pm 0.08$, and $0.81 \pm 0.09$ for 440, 550, 670, and 870 nm, respectively. Retrieved single scattering albedo values are well within the range measured during the FLAME 2 experiment using a photoacoustic spectrometer and a nephelometer and are close to the retrieved values from SPEX airborne and RSP by Fu et al. (2020). Fine- and coarse-mode AOD calculated using the log-normal modes listed in Table 3 are plotted in Fig. 10a and b. The main contribution to AOD is from the fine particles (log-normal modes 1 to 3 in Table 3), with almost no contribution from coarse mode (log-normal modes 4 and 5 in Table 3). The retrieved volume median radius for the fine mode ($r_{v,fine}$; see Eq. B1f) is $0.157 \pm 0.024\,\mu m$, with $\sigma_{v,fine} = 0.550 \pm 0.026$. Figure 11a shows the AE derived for each pixel calculated using the 440 and 870 nm spectral bands. Figure 11b and c are aerosol volume concentration ($\mu m^3\,\mu m^{-2}$) and SF retrieved, respectively, for each pixel inside the red box in Fig. 8. The Angstrom exponent

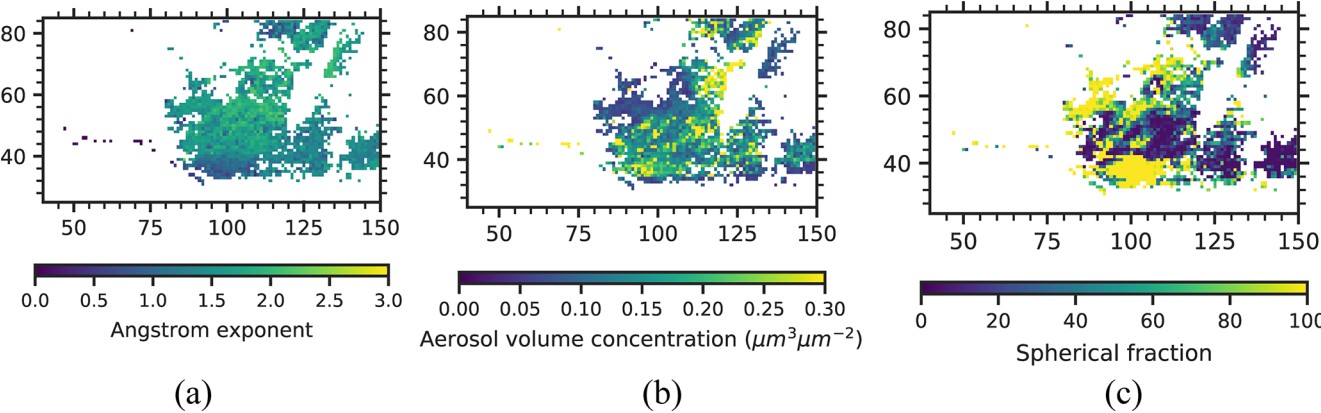

**Figure 11.** Map of **(a)** Angstrom exponent (AE), **(b)** aerosol volume concentration, and **(c)** spherical fraction (SF) of the scene shown in Fig. 9. Pixel filtering similar to the one applied for SSA, RRI, and IRI in Fig. 9 is applied to filter out bad pixels from the AE, aerosol volume concentration, and SF map. $X$ and $Y$ axes are the pixel coordinates.

retrieved from our measurement has a mean value of 1.53 with a standard deviation (SD) of 0.336. Also, we see a significant number of pixels with AE > 2, which is considered to be fresh smoke (Nicolae et al., 2013). In the smoke scene retrieved from AirHARP measurements the mean value of the percentage of spherical particles is 50 % with a SD of 36 %. It essentially means that the retrieved particle shapes of the smoke particles have been retrieved as highly nonspherical for much of the smoke plume, while reports from SPEX and RSP for the same smoke scene indicate the opposite, with 99 % and 85 % spherical for SPEX airborne and RSP, respectively (Fu et al., 2020). However, we see that in our retrievals for the scene in Fig. 8 there are a significant number of smoke pixels with spherical fractions close to 100 % (see Fig. 11c). While smoke properties are often spherical (Manfred et al., 2018; Martins et al., 1998), nonspherical fractal shapes can be seen in scanning electron microscopy (Chakrabarty et al., 2006). There is no definite answer whether the results shown in Fig. 11c are a retrieval artifact or are physically true. We do know from experimentation that the GRASP retrievals in this situation were not particularly sensitive to particle shape, returning the same values for AOD and SSA, within uncertainty bounds, whether SF was held constant at 80 %–99 % or whether it was a free parameter and retrieved as in Fig. 11c. A table of retrieved and derived parameters is listed in Table 6. In order to check the quality of the data fitting for the smoke scene of Figs. 8–11, 2-D density plots of measured and fitted variables, $R_I$ and DoLP, for all spectral bands are plotted in Fig. 12. Figure 12a, c, e, and g show the 2-D density plot for the reduced radiance $R_I$ measured at the four spectral bands 440, 550, 670, and 870 nm, respectively. Two-dimensional density plots of the measured and GRASP-fitted DoLP at 440, 550, 670, and 870 nm are plotted in Fig. 12b, d, f, and h, respectively. The fit for DoLP in 870 nm band has a higher spread in the density plot compared to the other spectral bands because the

silicon-based detector used for imaging in AirHARP has a lower quantum efficiency at 870 nm compared to the three other wavelengths. The 550 and 670 nm band data show the best correlation with GRASP fit with $R^2 = 0.991$ and 0.993 for $R_I$, whereas 440 and 870 nm have $R^2 = 0.986$ and 0.990, respectively. OLS regression for the 440 nm yields a slope of 0.984, which is the least-performing band for the variable $R_I$, followed by the 870, 550, and 670 nm spectral bands. For the case of DoLP, 870 nm has the lowest $R^2$ value of 0.960, followed by 670 nm ($R^2 = 0.991$). Both the 440 and 550 nm bands have an $R^2$ value of 0.995. Unlike the variable $R_I$, the DoLP in the 550 nm band shows more deviation from the 1 : 1 line, with a slope of 0.964 for OLS regression fit. Overall, the 2-D density plots reveal that the fitting for each variable $R_I$ and DoLP generated using the GRASP and AirHARP measurements performs well for the smoke scene in Fig. 8. Since the retrieval is an ill-posed mathematical problem, we need to make sure that the retrieved values are reasonable and compatible with co-incident instruments. For the case of the ACEPOL campaign, AODs from the AirHARP–GRASP retrievals are verified by comparing it with HSRL2 and AERONET observations. In the next subsection, we use the flight leg on 9 November 2017 at 19:30 UTC to compare the AOD retrievals from AirHARP with the collocated HSRL2 measurements.

## 5 Comparison of AirHARP GRASP retrievals with collocated data sets

### 5.1 High Spectral Resolution LIDAR-2 (HSRL2) vs. AirHARP AOD comparison

HSRL2, flying on the same aircraft with AirHARP during ACEPOL, provides the opportunity to compare the GRASP retrievals of AOD with an independent and collocated measurement. AirHARP lacks a wavelength channel identical

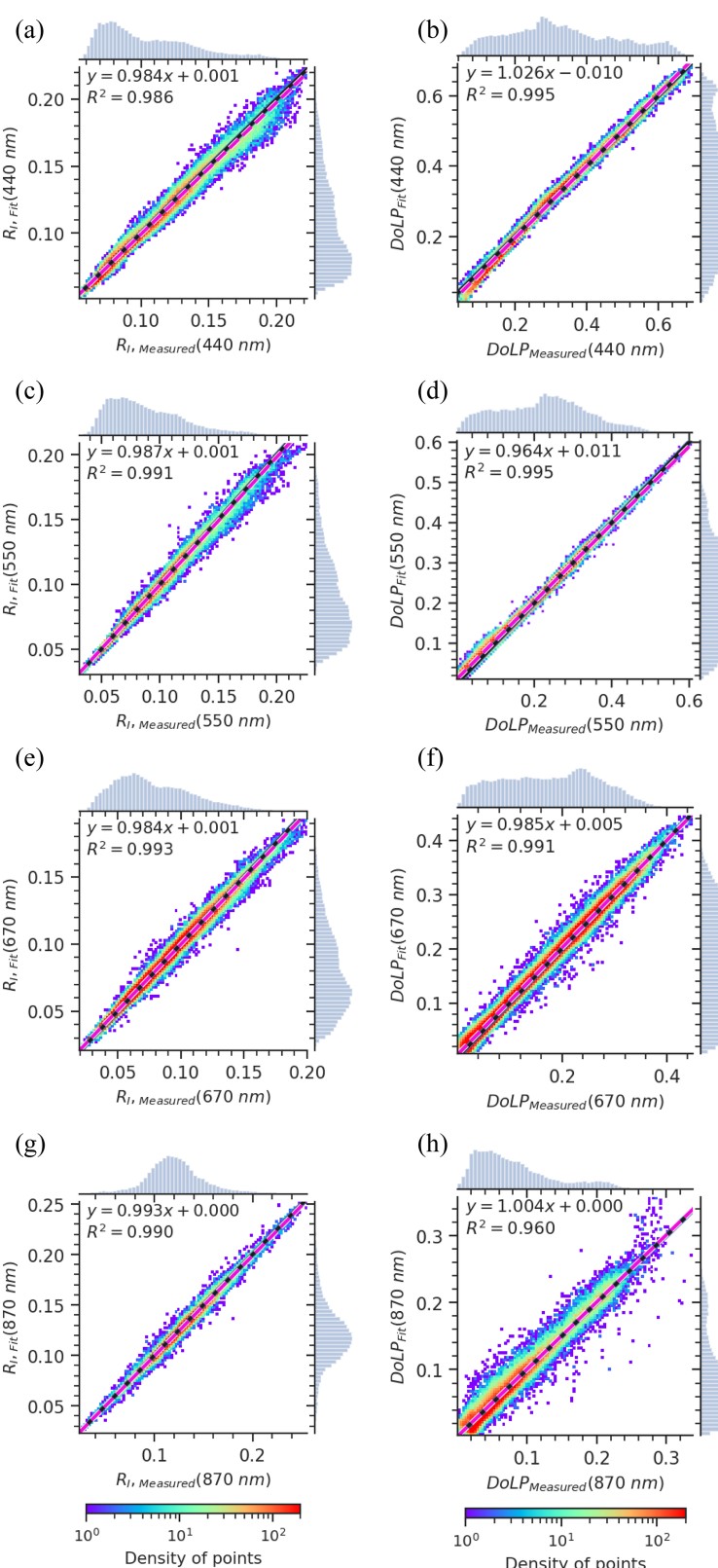

**Figure 12.** Scatter density plot for AirHARP measurement variables and GRASP fit for the scene in Fig. 9. 1 : 1 line (black solid line), OLS regression line (magenta dashed line), the OLS fit parameters and correlation ($R^2$) are also reported in the same graph. For each plot, the histogram of measurement and GRASP fit for $R_I$ and DoLP are plotted on the top and right axes, respectively.

to the wavelengths measured by HSRL2; therefore, for this study, we make use of the 440 and 550 nm channels on AirHARP to calculate the Angstrom exponent and then use that information to interpolate the AOD to HSRL2's wavelength of 532 nm. We collocate HSRL2 and AirHARP measures of AOD for the smoke plume shown in Fig. 4d (9 November 2017; 19:30 UTC). The smoke plume in this image is a controlled fire started in the Kaibab National Forest, Arizona, USA, and is highly non-homogenous near the source fire. The extreme non-homogeneity of the smoke in this scene introduces additional uncertainty to the particle property retrievals, and only AOD will be shown here. To match the HSRL2 ground pixel in the AirHARP image, the latitude and the longitude are matched to a tolerance level of about 200 m on the ground. For this flight leg, HSRL2 reports the aerosol extinction at 10 s intervals, which translates to 2 km in the ground distance for an ER-2 aircraft flying at 20 km altitude with a speed of $\sim 200 \, \mathrm{m \, s^{-1}}$ but with a narrow cross-track footprint of only 15 m. Thus the 2000 m × 15 m footprints of the HSRL2 measurements are inherently mismatched with AirHARP's 275 m × 275 m pixels. Given the inhomogeneity of this aerosol, we do not expect perfect agreement between the two sensors' retrieved AODs, simply because of the mismatch in spatial sampling.

For this study, we make use of the HSRL2 AOD at 532 nm, where an assumption of the lidar ratio is not required. All the GRASP retrievals for this comparison are done using the $R_I$ and DoLP measurements from AirHARP. Aerosol optical depth at 532 nm as a function of the collocated along-track pixels is plotted in Fig. 13a, and the scatter plot of the comparison is shown in Fig. 13b. In Fig. 13a HSRL2-measured AOD is denoted by the green diamond markers and AirHARP AOD by grey squares. Each square represents the mean of 28 pixels around the collocated HSRL2 ground pixel in the AirHARP image. We used seven pixels along-track ($\sim 1.93 \, \mathrm{km}$) and four pixels cross-track ($\sim 1.1 \, \mathrm{km}$) to find those mean values. The error bar in the AirHARP data points is the SD of AOD of all pixels within the $\sim 1.93 \, \mathrm{km} \times 1.1 \, \mathrm{km}$ region around the HSRL2 ground pixel, representing the spatial variability of the smoke plume within the averaging rectangular box. For this heterogenous smoke plume case, we had to apply $\chi^2_{\mathrm{norm}} < 20$, to filter out bad pixels/fits. Non-homogeneity of the smoke makes the retrieval complicated since we see different parts of the smoke plume when we scan through the different viewing angles of AirHARP data. So, using a higher $\chi^2_{\mathrm{norm}}$ value for filtration helps to catch the higher AOD values. A scatter plot of these two data sets is shown in Fig. 13b and the solid black line is the 1 : 1 line, whereas the yellow error bar represents the spatial variability of AirHARP AOD, similar to that in Fig. 13a. A comparison of HSRL2-measured AOD at 532 nm with the AirHARP AOD retrievals at 532 nm shows a strong, positive correlation and only deviates when the plume is thick and heterogeneous. A Pearson correlation coefficient ($\rho$) of 0.940, BIAS $= -0.062$, and mean absolute error (MAE) $= 0.122$ is

obtained for this comparison. Matching the HSRL2 AOD in regions of heterogeneity is challenging due to spatial mismatch between AirHARP and HSRL2 pixels. This will create issues where there is a sharp variation in the AOD, like close to the source and in the boundary of the smoke plume. The different cross-track pixel size between the HSRL2 and AirHARP measurements makes the intercomparison difficult to interpret in some cases. For points near the plume source, higher pixel variability may also bias AirHARP AOD retrievals performed at the same general location as the HSRL2 measurement. In a scene with this much complexity, there is additional uncertainty in matching multi-angle views for the AirHARP retrieval, because each viewing angle of the instrument will be looking at a different plume thickness, and this violates the plane-parallel assumption.

## 5.2 AERONET vs. AirHARP AOD comparison

Validation of AirHARP–GRASP retrievals using AERONET-measured aerosol optical depth for the collocated AirHARP pixels during the ACEPOL campaign are discussed in this section. A list of the collocated AERONET stations and measurements used for this analysis is given in Table 2. Only AERONET stations with a data quality of Level 2.0 are used for the comparison. For the AERONET validation, AirHARP pixels with a resolution of 550 m × 550 m are used for the GRASP retrievals to avoid the issues due to the small pixel shifting during the reprojection to a common latitude–longitude grid as well as to avoid strong fine-resolution surface features that appear over the urban area. To further protect the algorithm from subpixel inhomogeneity and other features inappropriate for retrieval, a $\chi^2_{\mathrm{norm}} < 1.5$ filter is used to remove the bad pixels/fits which may be caused by the presence of thin clouds (Stap et al., 2015) or due to the inability of surface reflection models to represent the directional reflectance from a complicated surface. To collocate the AERONET station (a single pixel) within the AirHARP image, the latitude and longitude of the AERONET location are matched to the AirHARP latitude and longitude with a tolerance of $2 \times 10^{-3\circ}$, which is approximately equivalent to 200 m on the ground. An area of 5.5 km × 5.5 km (10 × 10 retrieval pixels) around this collocated pixel is used for the calculation of the area mean AOD from the AirHARP retrievals, and this is matched to a 1 h temporal mean from AERONET measurements. Each of the 5.5 km × 5.5 km averaging boxes includes 100 pixels; however, many of them are removed after the $\chi^2_{\mathrm{norm}}$ filtering. AERONET-measured AOD is interpolated linearly in log–log space using the AE-to-AirHARP spectral bands for 1 : 1 comparison. The Ross–Li BRDF surface model along with the Maignan–Breon BPDF models are used for representing the directional reflectance from the land surface for all retrievals used in the validation.

AirHARP observations in this validation exercise are retrieved with two versions of the GRASP aerosol kernels, one

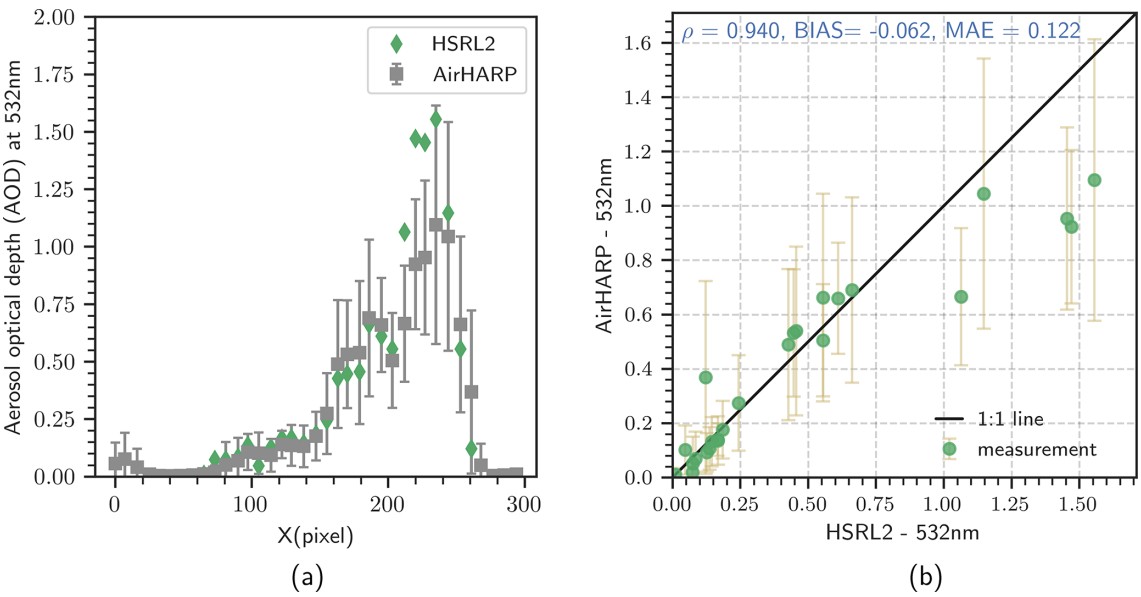

**Figure 13. (a)** AOD at 532 nm from AirHARP vs. HSRL2 AOD at 532 nm along the flight track for the forest fire scene on 9 November 2017, T19:30 UTC; **(b)** correlation plot for the HSRL2 AOD at 532 nm vs. AirHARP AOD at 532 nm for the same flight.

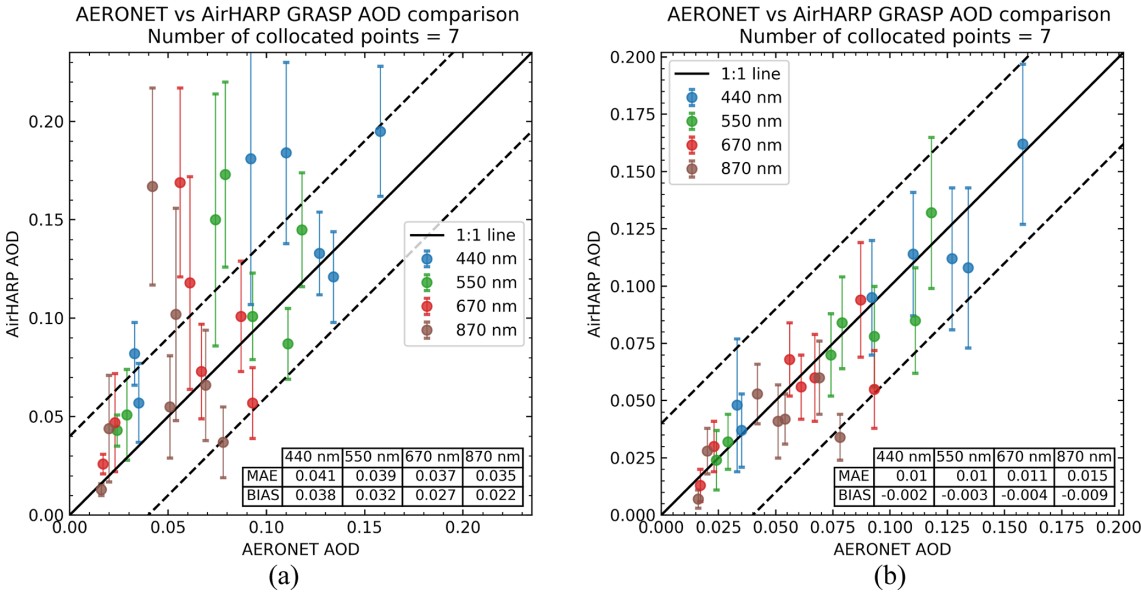

**Figure 14.** Scattergrams of aerosol optical depth (AOD) retrieved using AirHARP observations over collocated AERONET stations vs. AERONET-measured AOD interpolated to AirHARP spectral bands. Plotted are the areal mean AirHARP AODs calculated from all the qualified retrievals within a box of 5.5 km × 5.5 km around the AERONET station against the collocated AERONET temporal mean AOD. Each colored error bar indicates the SD of AOD within the areal box. Mean absolute error (MAE) and BIAS for each spectral band are provided in the table inside the scattergram. The black solid lines in the plots are the 1 : 1 lines, and the dashed lines are ± 0.04 AOD from the 1 : 1 line. **(a)** GRASP retrievals using the GRASP/Five mode kernel (Table 3) that has 15 free parameters and allows for retrieval of particle properties along with AOD; **(b)** same as **(a)** except using the GRASP/Models kernel (Table 4) that reduces free parameters to six, fixes particle properties based on Table 4 and only retrieves the concentration for each aerosol component.

using the GRASP/Five mode kernel with 15 free parameters (Table 3) that allows for retrieval of particle properties and the other using the GRASP/Models kernel with only 6 free parameters (Table 4) that restricts the particle properties to focus on the AOD retrieval. When aerosol loading is low, there is insufficient signal to retrieve particle properties. Allowing for additional free parameters without having sufficient signal will degrade the accuracy of the AOD retrieval.

The maximum AOD measured by a collocated AERONET station during the ACEPOL campaign is 0.158 at 440 nm, suggesting that in this exercise the simplified aerosol component model would be preferred over using the option with a greater number of free parameters. This is evident in the two figures, Fig. 14a and b. Figure 14 shows the scatter plot with AOD$_{AERONET}$ on the $X$ axis and AOD$_{AirHARP}$ on the $Y$ axis. The spatial SD of AOD within this 5.5 km × 5.5 km box is indicated using the error bars in Fig. 14. Statistical parameters that represent the correlation between these two data sets are shown in the table inside the plots. In Fig. 14a, for the case of AOD retrievals based on the GRASP/Five mode kernel with the greater number of free parameters, MAEs of 0.041, 0.039, 0.037, and 0.035 are obtained for the 440, 550, 670, and 870 nm bands, respectively. The bias ranges from 0.022 to 0.038, with the 440 nm band having the largest bias and the 870 nm band with the least. However, in Fig. 14b, for the case using the GRASP/Models kernel with the reduced number of free parameters, mean absolute errors of 0.010, 0.010, 0.011, and 0.015 are obtained for the spectral bands 440, 550, 670, and 870 nm, respectively, with the 870 nm band having a slightly higher spread than the other bands. Also, a similar trend is seen for 870 nm in the case of BIAS, where the 440, 550, and 670 nm bands have a BIAS of −0.002, −0.003, and −0.004, respectively, whereas 870 nm has a BIAS of −0.009. Figure 14 demonstrates the need to match the appropriate kernel to the available information in the scene and to not attempt the retrieval of more free parameters than the aerosol loading permits. Overall, the performance of the AirHARP observations plus the GRASP retrieval algorithm gives a good agreement with the collocated AERONET observations, especially when the GRASP/Models kernel is used. The above tendency is well identified in the in-depth analysis of PARASOL data processing using different retrieval setups by Chen et al. (2020).

## 6  Future research

We note the abundance of very low aerosol loading in the majority of the ACEPOL flight legs. Future work should make use of these data to focus on surface characterization using the AirHARP–GRASP combination. There are several flight segments over the Rosamond Dry Lake, in California, USA, in different flight directions, and because the dry lake is relatively flat and uniform, it becomes an ideal target for testing GRASP retrievals of surface BRDF and BPDF parameters. Furthermore, on this day, at AirHARP overpass, ground measurements of radiance and polarized radiance were made using the Ground Multiangle Spectro-Polarimetric Imager. This will be a perfect case for a case study on the performance of different BRDF + BPDF kernel combinations in representing the directional reflectance from a bright surface. This data set can be used to improve the BRDF/BPDF model at an unprecedented higher resolution compared to previous studies (Maignan et al., 2004, 2009). This will help in better characterization of directional reflectance from the urban surface and will benefit the overall accuracy of AOD retrievals over the land.

## 7  Conclusions

In this study, AirHARP polarimetric measurements, taken at the high angular and spatial resolution over a wide swath, combined with the GRASP algorithm allow for unprecedented spatial mapping of aerosol properties that are consistent with co-incident instrument retrievals. These properties always include AOD but can also deliver real and imaginary parts of the refractive index, particle size information, spherical fraction, and single scattering albedo when the aerosol is suitably homogeneous and loading is sufficiently high. We demonstrated that the measurements match the forward model calculations in a variety of environments: the retrieval performs well over barren land surface and vegetation, though retrievals over sunglint still show biases in the AOD. The wide swath of the AirHARP enables the aerosol retrievals over a large scene of interest, which makes the AirHARP instrument unique from many of the other airborne polarimeters available as of today. Also, this wide swath will help in capturing more aerosol events globally compared to a narrow-swath multi-angle polarimeter when the HARP concept is applied to space sensors.

In situations with low aerosol loading (AOD < 0.17) over land, a simplified retrieval approach based on GRASP/Models kernel approximating aerosol as an external mixture of five aerosol components is also used for the AOD retrievals. One advantage of using this simplified kernel is that it retrieves a significantly smaller number of aerosol parameters compared to the standard GRASP/Five mode kernel and performed well for low aerosol loading cases in an AERONET comparison, despite the simplifying assumption of a prescribed complex refractive index for each aerosol component. AOD retrieved from AirHARP using GRASP matches collocated AERONET observations to within +0.018/−0.04 with a minimum MAE of 0.01 in the 440 and 550 nm bands and a maximum of 0.015 in the 870 nm spectral band. Thus, we note an overall low bias of −0.002 to −0.009, depending on wavelength. Traditionally, low AOD conditions over land were some of the most difficult situations for standard operational aerosol retrieval algorithms applied to orbiting radiometers. For example, the MODIS Dark Target algorithm reports an accuracy at a low loading of ± 0.05 (Levy et al., 2013), MODIS Deep Blue ± 0.03 (Sayer et al., 2013), and MISR ± 0.03 to ± 0.05 (Kahn et al., 2010). All of these radiometer products expect twice the uncertainty at the low loading end than what was obtained from AirHARP/GRASP in these circumstances. Granted that these results were achieved for only seven AirHARP overpasses of AERONET sites and will need

to be reproduced in a variety of settings and situations, but for now, the match-ups with AERONET are very promising for AOD retrievals over complex land surfaces.

GRASP was applied to AirHARP observations of two heavy aerosol loading situations, both of smoke plumes near fire sources. It is in one of these situations that optical and microphysical characteristics of the smoke, in addition to AOD, are retrieved using the GRASP software. Retrievals of smoke properties on 27 October 2017 T18:16 UTC show that particles are fine with the real and imaginary refractive indexes of 1.55 and 0.024, respectively. Single scattering albedos of 0.87, 0.86, 0.84, and 0.81 are retrieved for the smoke at 440, 550, 670, and 870 nm, respectively, with a fine-mode volume median radius of 0.157 μm and SD of 0.55 with a mixture of spherical and non-spherical particles. The isolated location of these local smoke plumes prevents any validation with AERONET observations. However, the retrieved optical and microphysical properties fall within expectations of fire smoke from previous publications, including a report for a similar smoke from two other ACEPOL polarimeters (RSP and SPEX airborne) flying on the same aircraft as AirHARP. See the discussion in Sect. 5. Furthermore, the AirHARP-/GRASP-retrieved AOD agrees well with collocated HSRL2 observations of the smoke plume to the degree to which the two sensors with very different observation geometries and cross-track pixel sizes can be matched. Note that when the plume is highly heterogeneous, the smaller cross-track footprint of HSRL2 measurements relative to AirHARP makes collocation extremely difficult. Also, the complex structure of the plume violates the plane-parallel aerosol layer assumption of the retrieval, adding uncertainty and bias.

AirHARP is the first manifestation of the HARP concept that makes multi-wavelength, hyper-angle, polarization measurements across a wide swath. Thus, the encouraging results demonstrated here show that when combined with GRASP inversion methods, HARP measurements have the potential to be used to retrieve accurate measures of AOD. With sufficient aerosol loading and homogeneity, aerosol optical/microphysical characteristics can be retrieved over a broad area. HARP CubeSat that will fly at the International Space Station orbit of ∼410 km in 2020 will cover ∼1348 km in its across-track image, while HARP2 that will be part of the Plankton, Aerosol, Clouds, ocean Ecosystem (PACE) mission at ∼676 km orbit will image ∼1550 km across its swath. This AirHARP/GRASP demonstration encourages high expectations for these future HARP space missions. Furthermore, HARP has the potential to provide new characterization for clouds (McBride et al., 2020) and surface properties over various surface types, land and ocean. The PACE mission also opens the corridor for a synergetic observation using the Ocean Color Instrument (OCI) along with the two multi-angle polarimeters: HARP2 and SPEXone. OCI is a hyperspectral, wide-swath radiometer, HARP2 a wide-swath multi-angle polarimeter, and SPEXone a hyperspectral narrow-swath multi-angle polarimeter.

The combined spectral and spatial coverage and resolution of these three instruments will provide an unprecedented data set for the atmospheric, ocean, and terrestrial science research communities (Frouin et al., 2019; Hasekamp et al., 2019; Remer et al., 2019). These new capabilities for aerosol, cloud, and surface characterization will offer the community new insight into important climate processes, public health issues, and other societal concerns. TS7

## Appendix A

### A1 Land surface models

GRASP includes multiple land and ocean surface reflectance models. Here we will be discussing only the one we have used for the retrievals in this paper. Land reflectance BRDF and BPDF models are derived using an analytical (Cook et al., 2004; Rahman et al., 1993; Roujean et al., 1992; Wanner et al., 1995) and semi-empirical concept (Breon and Maignan, 2017), which used the heritage data from MODIS and POLDER to characterize the BRDF and BPDF models with higher accuracy. Currently, we use two land models for the retrievals.

#### A1.1 Ross–Li BRDF model

This model characterizes the directional reflectance from the surface which is illuminated from a direction $(\vartheta_1, \phi_1)$ and observed from a direction $(\vartheta_2, \phi_2)$ by the linear combination of three types of scattering kernels and is given by

$$
\begin{aligned}
\rho_{\text{Ross-Li}}&(\vartheta_1, \phi_1, \vartheta_2, \phi_2) \\
&= f_{\text{iso}}(\lambda) + K_{\text{vol}}(\lambda) f_{\text{vol}}(\vartheta_1, \phi_1, \vartheta_2, \phi_2) \\
&+ K_{\text{geo}}(\lambda) f_{\text{geo}}(\vartheta_1, \phi_1, \vartheta_2, \phi_2).
\end{aligned}
\tag{A1a}
$$

$f_{\text{iso}}$, $f_{\text{vol}}$, and $f_{\text{geo}}$ are the three kernels which represent isotropic, volumetric, and geometric-optical surface scattering, respectively. Litvinov et al. (2011) show that surface reflectance can be represented as a product of geometrical and wavelength-dependent terms.

$$
R_I(\vartheta_1, \phi_1, \vartheta_2, \phi_2, \lambda) \approx k(\lambda) f_i(\vartheta_1, \phi_1, \vartheta_2, \phi_2)
\tag{A1b}
$$

From Eqs. (A1a, A1b) we can derive $R$ as

$$
\begin{aligned}
R_I&(\vartheta_1, \phi_1, \vartheta_2, \phi_2, \lambda) \\
&= k(\lambda)\big[1 + k_1 f_{\text{geo}}(\vartheta_1, \phi_1, \vartheta_2, \phi_2) \\
&+ k_2 f_{\text{vol}}(\vartheta_1, \phi_1, \vartheta_2, \phi_2)\big],
\end{aligned}
\tag{A1c}
$$

where $k_1$ and $k_2$ are wavelength-independent linear model parameters for geometrical-optical and volumetric surface scattering kernels, respectively. The $k(\lambda)$ is a wavelength-dependent model parameter. The volumetric kernel is defined as (Ross, 1981; Roujean et al., 1992; Wanner et al., 1995)

$$
f_{\text{vol}}(\vartheta_1, \phi_1, \vartheta_2, \phi_2) = \frac{\left(\frac{\pi}{2} - \gamma\right)\cos\gamma + \sin\gamma}{\cos\vartheta_1 + \cos\vartheta_2} - \frac{\pi}{4},
\tag{A1d}
$$

where $\gamma$ is the scattering defined in the scattering plane and is given by the equation

$$
\gamma = \cos^{-1}\left(-\cos\vartheta_1 \cos\vartheta_2 - \sin|\vartheta_2|\sin\vartheta_1 \cos\phi\right).
\tag{A1e}
$$

Here, $\phi = \phi_2 - \phi_1$ is the relative azimuthal angle. The Li-sparse geometric kernel is defined as

$$
\begin{aligned}
f_{\text{geo}}&(\vartheta_1, \phi_1, \vartheta_2, \phi_2) \\
&= O(\vartheta_1, \phi_1, \vartheta_2, \phi_2) - \sec\vartheta'_1 - \sec\vartheta'_2 \\
&+ \frac{1}{2}(1 - \cos\vartheta')\sec\vartheta'_1 \sec\vartheta'_2,
\end{aligned}
\tag{A1f}
$$

$$
O = \frac{1}{\pi}(t - \sin t \cos t)\left(\sec\vartheta'_1 + \sec\vartheta'_2\right),
\tag{A1g}
$$

$$
\cos t = \frac{h}{b}\frac{\sqrt{D^2 + (\tan\vartheta'_1\tan\vartheta'_2 \sin\phi)^2}}{\sec\vartheta'_1 + \sec\vartheta'_2}
\tag{A1h}
$$

$$
D = \sqrt{\tan^2\vartheta'_1 + \tan^2\vartheta'_2 + 2\tan\vartheta'_1\tan\vartheta'_2 \cos\phi},
\tag{A1i}
$$

$$
\cos\vartheta' = \cos\vartheta'_1 \cos\vartheta'_2 - \sin\vartheta'_1 \sin\vartheta'_2 \cos\phi,
\tag{A1j}
$$

$$
\vartheta'_1 = \tan^{-1}\left(\frac{b}{r}\tan\vartheta_1\right), \quad \vartheta'_2 = \tan^{-1}\left(\frac{b}{r}\tan|\vartheta_2|\right).
\tag{A1k}
$$

To reduce the number of surface parameters retrieved, in linear models $h/b$ and $b/r$ are fixed. Like the MODIS BRDF retrieval algorithm (Schaaf et al., 2002), we predefine the values $h/b = 2$ and $b/r = 1$ in the BRDF kernel.

#### A1.2 Maignan–Breon BPDF model

Most of the theoretical models developed for the BPDF are based on the Fresnel equation of light reflection from a surface. The Nadal–Breon model uses a two-parameter nonlinear Fresnel function to characterize the aerosol over land polarized reflectance. The polarized reflectance can be given by

$$
R_{\text{P}}(\vartheta_1, \phi_1, \vartheta_2, \phi_2) = \alpha\left(1 - \exp\left(-\beta\frac{F_{\text{p}}(m, \gamma)}{\cos\vartheta_1 + \cos\vartheta_2}\right)\right),
\tag{A1l}
$$

where $F_{\text{p}}$ is the polarized Fresnel reflection coefficient and is given by

$$
F_{\text{p}} = \frac{1}{2}\left(r_\perp^2 + r_\parallel^2\right),
\tag{A1m}
$$

where $r_\perp$ and $r_\parallel$ are perpendicular and parallel components of Fresnel reflection coefficients, respectively. Maignan et al. (2009) implemented a simple one-parameter model compared to the complicated two-parameter Nadal–Breon model to represent the polarized reflectance from aerosol over the land surface. It has been used in the POLDER/PARASOL retrieval algorithm as a primary model for the BPDF over land. The one-parameter BPDF model is given by

$$
R_{\text{P}}(\vartheta_1, \phi_1, \vartheta_2, \phi_2) = \frac{\alpha\exp(-\tan\theta_r)F_{\text{p}}(m, \gamma)}{4(\cos\vartheta_1 + \cos\vartheta_2)}.
\tag{A1n}
$$

Here $\alpha$ is the only free linear parameter. $\theta_{\text{r}}$ is the angle of specular reflection.

## A2 Ocean surface models

For the surface model in the RT calculation, we used a modified Cox–Munk isotropic model. A detailed description of this can be found in the references (Kawata et al., 1995; Mishchenko and Travis, 1997; Sun and Lukashin, 2013). The modified Cox–Munk model calculates the BRDF and BPDF based on three parameters, and $a_0$ is the albedo of the ocean surface, which is spectrally dependent and smooth. $a_1$ is the fraction of Fresnel's reflection surface and $a_2$ is the variance of wind speed distribution. Wind speed distribution is given by the equation

$$p(Z_x, Z_y) = \frac{1}{\pi a_2} \exp\left(-\frac{Z_x^2 + Z_y^2}{a_2^2}\right), \tag{A1o}$$

where

$$a_2^2 = 0.003 + 0.00512\,V. \tag{A1p}$$

$Z_x$, $Z_y$ are $X$ and $Y$ slope components. $V$ is the surface wind speed in $\mathrm{m\,s^{-1}}$.

## Appendix B

### B1 Aerosol optical depth (AOD) calculation

In GRASP, atmospheric aerosol particles are considered a mixture of spherical and randomly oriented spheroid particles, and the aerosol optical depth is modeled as follows:

$$\tau_{\mathrm{aero}}(\lambda) = \tau_{\mathrm{spherical}}(\lambda) + \tau_{\mathrm{spheroid}}(\lambda) \tag{B1a}$$

$$\tau_{\mathrm{spherical}}(\lambda) = \int\limits_{\ln r_{\min}}^{\ln r_{\max}} \frac{C_{\mathrm{ext}}^{\mathrm{sphere}}(\lambda, n, k, r)}{v(r)} \frac{\mathrm{d}V(r)}{\mathrm{d}\ln(r)} \mathrm{d}(\ln(r)), \tag{B1b}$$

and

$$\tau_{\mathrm{spheroid}}(\lambda) = \int\limits_{\ln \varepsilon_{\min}}^{\ln \varepsilon_{\max}} \int\limits_{\ln r_{\min}}^{\ln r_{\max}} \frac{C_{\mathrm{ext}}^{\varepsilon}(\lambda, n, k, r)}{v(r)} \frac{\mathrm{d}n(\varepsilon)}{\mathrm{d}\ln(\varepsilon)}$$

$$\frac{\mathrm{d}V(r)}{\mathrm{d}\ln(r)} \cdot \mathrm{d}(\ln(r))\mathrm{d}(\ln(\varepsilon)), \tag{B1c}$$

where $\lambda$ is the wavelength, $n$ is the real part of the refractive index, $k$ is the imaginary part of refractive index, $v(r)$ is the volume of particles, $C_{\mathrm{ext}}^{\mathrm{sphere}}$ and $C_{\mathrm{ext}}^{\varepsilon}$ are extinction cross sections of spherical and spheroid particles with aspect ratio $\varepsilon(\varepsilon = \frac{a}{b}$, $a$ is the axis of spheroid rotational symmetry, and $b$ is the axis perpendicular to the axis of spheroid rotational symmetry), respectively. $\frac{\mathrm{d}n(\varepsilon)}{\mathrm{d}\ln(\varepsilon)}$ used in GRASP is a fixed shape distribution as mentioned in Dubovik et al. (2006). Integrals in Eqs. (B1b) and (B1c) are changed to sum in order to do the computation fast and accurate, and thus it becomes

$$\tau_{\mathrm{spherical}}(\lambda) = \sum_{i=1}^{N_r} \left(\mathrm{SF} \cdot K_{\tau}^{\mathrm{spherical}}(\lambda, n, k, r)\right) \frac{\mathrm{d}V(r_i)}{\mathrm{d}\ln(r)}, \tag{B1d}$$

where SF is the spherical fraction and $K$ is the quadrature coefficient of extinction and is pre-computed kernels. Precomputed $K$ has been calculated for a wide range of $n$ ($1.33 \leq n \leq 1.7$) and $k$ ($0.0005 \leq k \leq 0.5$). The calculations were done for a fixed aspect ratio from 0.3 to 3.0 and narrow size bins cover the size parameter range from $\sim 0.0012$ to $\sim 625$.

$$\tau_{\mathrm{spheroid}}(\lambda) = \sum_{i=1}^{N_r} (1 - \mathrm{SF}) K_{\tau}^{\mathrm{spheroid}} \frac{\mathrm{d}V(r_i)}{\mathrm{d}\ln(r)} \tag{B1e}$$

Also,

$$\frac{\mathrm{d}V_i(r)}{\mathrm{d}\ln(r)} = \sum_{i=1,\ldots,N} \frac{C_{v,i}}{\sqrt{2\pi}\sigma_i} \exp\left(\frac{-(\ln r - \ln r_{v,i})^2}{2\sigma_i^2}\right). \tag{B1f}$$

$C_{v,i}$ is the concentration of different modes, $r_{v,i}$ is the volume median radius of each mode, and $\sigma_i$ is the SD.

*Data availability.* The AirHARP L1B and HSRL2 data for the ACEPOL campaign can be accessed from the website https://www-air.larc.nasa.gov/cgi-bin/ArcView/acepol TS8. Quick look images for each of the AirHARP flight legs can be accessed through the website https://sites.google.com/view/airharp-acepol/home TS9. AERONET data used for this study can be downloaded from https://aeronet.gsfc.nasa.gov/ TS10.

*Author contributions.* AP designed and implemented the study and analysis. AP wrote the manuscript with significant contributions from LR. BM, HM, and VM managed the operation of the AirHARP instrument during the ACEPOL campaign. VM, BM, XX, and HB calibrated and developed a scheme to generate the L1B data, which are the input for this data analysis. OD and PL helped with the retrievals by providing the GRASP software and support. SB provided input for the HSRL2–AirHARP AOD comparison. All the authors contributed to the improvement of the manuscript with valuable comments and suggestions.

*Competing interests.* The authors declare that they have no conflict of interest.

*Acknowledgements.* We thank the ACEPOL organizing team for supporting the involvement of the AirHARP instrument in the project. We thank the PIs of the AERONET stations used in this study for maintaining and providing easy access to data. We also acknowledge the engineering contributions from Roberto Fernadez-Borda (ESI, UMBC) and Dominik Cieslak (ESI, UMBC) in designing and manufacturing the AirHARP instrument. We acknowledge the former Aerosol, Cloud, Ecosystem (ACE) program at NASA's Earth Science Division as a sponsor for ACEPOL flights. The Dutch contribution to the ACEPOL flight campaign hours was funded by the SRON Netherlands Institute for SPACE Research and by NSO/NWO under project number ALW-GO/16-09. The authors acknowledge the support of Anton Lopatin (GRASP-SAS) and Tatsiana Lapionak (LOA Université Lille) in providing the GRASP kernels for this study. The authors acknowledge the work by the Python Software Foundation (http://www.python.org, last access: 8 September 2020) and Anaconda Inc. (https://www.anaconda.com/, last access: 8 September 2020) in developing and maintaining the Python programming language and packages used for analyzing and visualizing the data used in this study.

*Financial support.* This research is primarily supported through NASA projects HARP CubeSat (grant no. NNX13AN11G), AirHARP (grant no. NNX16AK36G), and JCET cooperative agreement (grant no. NNX15AT34A). This research is also supported by the National Key R&D Program of China (grant no. 2016YFE0201400). Oleg Dubovik appreciates the support from the Chemical and Physical Properties of the Atmosphere Project funded by the French National Research Agency through the Programme d'Investissement d'Avenir (contract no. ANR-11-LABX-0 0 05-01), the Regional Council "Hauts-de-France", and the "European Funds for Regional Economic Development". Henrique M. J. Barbosa is grateful for the support of processo no. 2016/18866-

2 TS11, Fundação de Amparo à Pesquisa do Estado de São Paulo (FAPESP), and of project 308682/2017-3 TS12, Conselho Nacional de Desenvolvimento Cien tífico e Tecnológico (CNPq).

*Review statement.* This paper was edited by Sebastian Schmidt and reviewed by Otto Hasekamp and one anonymous referee.

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

## Remarks from the language copy-editor

CE1  Please give an explanation of why this needs to be changed. We have to ask the handling editor for approval. Thanks.
CE2  Please give an explanation of why this needs to be changed. We have to ask the handling editor for approval. Thanks.
CE3  Please give an explanation of why this needs to be changed. We have to ask the handling editor for approval. Thanks.
CE4  Please give an explanation of why this needs to be changed. We have to ask the handling editor for approval. Thanks.

## Remarks from the typesetter

TS1  Please confirm.
TS2  Please check throughout the text that all vectors are denoted by bold italics and matrices by bold roman.
TS3  Please note we use the letters for subequations.
TS4  Please note that there is no Eq. (2b) in the text; therefore, we use only one number.
TS5  Should be the subscripts of the Delta variables also vectors?
TS6  Please note that the matrix G has been deleted from this equation in accordance with the one provided in your corrections.
TS7  An appendix always starts on a new page, therefore the space.
TS8  Please provide a reference list entry including creators, title, and date of last access.
TS9  Please provide a reference list entry including creators, title, and date of last access.
TS10  Please provide a reference list entry including creators, title, and date of last access.
TS11  Does this project no. belong to the FAPESP?
TS12  Does this project no. belong to the CNPq?
TS13  Please notice. AMT update inserted.