# Peer review of "Retrieval of aerosol properties from Airborne Hyper Angular Rainbow Polarimeter (AirHARP) observations during ACEPOL 2017"

_Atmospheric Measurement Techniques, 2020_

## Referee Comment (RC1) · Otto Hasekamp (Referee) · 21 Apr 2020

This manuscript describes the first aerosol retrieval results from the airHARP instrument. The retrieval approach is being discussed and the algorithm is applied to a number of selected scenes from the ACEPOL campaign, where the fit quality between forward model and measurements is discussed as well as the comparison to independent measurements (HSRL-2 and AERONET). The paper is very relevant for the upcoming NASA PACE mission where similar (or improved) measurements and retrievals are to be expected. Overall, the comparison to the independent measurements looks good and some of the discrepancies (with HSRL2) can be explained. I would like

to congratulate the authors with these nice results.

My comments are included in the attached pdf. Most of them are minor, with the exception of one comment that I copy below:

It seems that 3 different GRASP setups are used for 3 different cases: - GRASP for fixed aerosol models, together with the RPV surface model for the AERONET comparison. - Retrievals for 5 aerosol modes with fixed refractive index, together with the Ross-Li surface model for the Rosamond Dry lake. - 'Full" retrievals for 5 aerosol modes where also microphysical properties are retrieved for the smoke case. Here also the Ross-Li surface model has been used.

These 3 setups should be clearly described in the Theory section and not in the Result sections where they are being used. Also, the choice for a given setup should be better motivated, and perhaps also some numbers for retrievals in other setups should be quoted.

If possible, I would even suggest to restrict to 2 setups: the ones corresponding to 'full' 'and models', and using the same surface model for all over land cases.

Also, I would like to encourage the authors to also quote the mean aerosol properties for the smoke plume on 9 November so that they can be compared with those retrieved by Fu et al. (AMT, 2020) for SPEX airborne and RSP. Further, for this day, the sphericity retrievals can be checked with the depolarization ratio from HSRL2.

Please also note the supplement to this comment:
https://www.atmos-meas-tech-discuss.net/amt-2020-64/amt-2020-64-RC1-supplement.pdf

---

## Referee Comment (RC2) · Anonymous Referee #2 · 12 May 2020

The paper by Puthukkudy et al. applied the GRASP algorithm to retrieve the aerosol properties from AirHARP observations. Following a) an in-depth overview of the GRASP inversion formalism and aerosol/surface models embedded in the inversion and b) an introduction of AirHARP instrument, retrieval tests were carried out using some ACEPOL campaign data AirHARP acquired during the ACEPOL 2017 campaign. Comparison to AERONET reference data via case study shows mean absolute error within 0.017. The paper is generally carefully written and technically correct. I have the following comments for the authors to further consider as improvements:

1. The authors review the airborne polarimeters (SPEX, RSP, AirMSPI). It is also nec-

essary to review aerosol retrieval algorithms used by these sensors and published elsewhere.

2. Associated with the comparison of AirHARP AOD results against AERONET reference data in Figure 14, how does the comparison of single scattering albedo and nonspherical particle fraction look like ? Though the AOD loading is low, are we still able to see a trend of improved agreement of these properties as AOD increases.

3. The authors need to be more clear on the adoption of size components in AirHARP retrieval. Table 3 and Table 5 give two different size components in retrieval. If I understand well, size components in Table 3 is default option in GRASP while size components in Table 6 was created for AirHARP retrieval. What the difference in the retrieval based on these two different assumptions ? Which one gives better fit to AERONET AOD and SSA ? As the authors pointed out "This simplified approach significantly drops the complexity of the aerosol model by reducing the number of parameters retrieved in the joint retrieval. It helps in reducing the nonlinearity of the inverse problem and makes the separation of the surface and aerosol signal much less complicated compared to a five-lognormal mode kernel." However, would it also possible that the size components in Table 6 (with fixed refractive index for eat mode) also increase the risk of getting more subjection to pre-determined aerosol modeling errors ? If so, how large is the modeling error ?

4. As the AOD loading during ACEPOL field campaign is low, it provides a very good testbed for surface retrieval. Is there any comparison of surface BRDF and pBRDF as retrieved from AirHARP and other sensors such as SPEX, RSP and AirMSPI ?

---

## Author Comment (AC1) · 23 Jun 2020

**Reply to Reviewer #1 comments**

The authors express sincere thanks to Dr. Otto Hasekamp for reviewing this research paper promptly with constructive comments and suggestions. We especially appreciate the suggestion of minimizing the number of GRASP configurations used. We now are down to one representation of land surface reflectance (Ross-Li BRDF) and two aerosol kernels. We primarily use the GRASP/Five mode kernel that allows for 15 free parameters but also use a GRASP/Models kernel with only 6 free parameters for a specific validation exercise when the AOD is low. We introduce a new figure that shows the difference in agreement with AERONET when using the different kernels in that validation exercise. For the ease of cross reading, authors responses are given under the reproduced version of each comment by Dr. Otto Hasekamp. Each of the comments from the reviewer are numbered and the changes made in the manuscript are shown using a red font under the authors response. The line numbers given in parenthesis at the end of author's comments will help the readers to locate the comments on the original manuscript submitted for discussion in AMT.

**Minor Comments**

1. *Comment on the aerosol volume concentration (Line #21)*

Changed aerosol volume concentration to aerosol column concentration.

The retrieved aerosol properties include spherical fraction (SF), aerosol  column concentration in multiple size distribution modes, and with sufficient aerosol loading, complex aerosol refractive index.

2. *Recommended to add the Mean absolute error for the HSRL2 AOD comparison (Line #25)*

Added the MAE and discussed the non-agreement caused by the edge of the plume, and the mismatch caused by the spatial sampling near the smoke source.

A good agreement with HSRL2 ($\rho = 0.940$, $—BIAS— = 0.062$, Mean Absolute Error ($MAE$) = 0.122) and AERONET AOD (0.010 ≤ $MAE$ ≤ 0.015, 0.002 ≤ $—BIAS—$ ≤ 0.009) measurements is observed for the collocated points. There was a mismatch between the HSRL2 and AirHARP retrieved AOD for the pixels close to the forest fire smoke source and to the edges of the plume due to spatial mismatch in the sampling. This resulted in a higher BIAS and MAE for the HSRL2 AOD comparison. For the case of AERONET AOD comparison two different approaches are used in the GRASP retrievals, and the simplified aerosol type-based kernel which retrieves fewer number of aerosol parameter performed well compared to a more generous approach in the low aerosol loading cases. Forest fire smoke intercepted during ACEPOL provided a situation with homogenous plume and sufficient aerosol loading to retrieve the real part of the refractive index (RRI) of 1.55 and the imaginary part of the refractive index (IRI) of 0.024.

3. *Recommendation on rephrasing (Line #36)*

Rephrased the sentence in the introduction.

they directly perturb Earth's radiation budget and indirectly by modifying cloud properties

 4. *Add the additional reference Hasekamp et. al 2011 for the POLDER AOD retrievals (Line #73)*

Added the reference

This capability has been demonstrated by space-borne POLDER I, II and III (POLarization and Directionality of the Earth's Reflectance) (Deuzé et al., 1999, 2001; Dubovik et al., 2019; Goloub et al., 1999; Hasekamp et al., 2011; Leroy et al., 1997),

 5. *Recommended to move a sentence into the theory section (Line #93-95)*

The authors prefer to keep this information in appendices and not move them to the theory section. These pages are meant as background information and rigorous completeness. They do not serve the casual reader if they appear in the flow of the main text.

 6. *Commented on the unconventional naming of Section 2.1 (Line #97)*

Yes, this is a bit unconventional but conveys the meaning properly.

 7. *Recommended using a more common notation for the reflectance calculated using the Stokes vector (Line #109, 113).*

Recommended change has been implemented in the manuscript, in its entirety, to use a more common notation for the reflectance.

 8. *Reviewer instructed to provide the uncertainty used in retrievals for the 870 nm channel (Line #144-145)*

The uncertainty/noise level used for the 870 nm band has been added to the manuscript.

The study in this paper uses 3 % radiometric uncertainty for all the bands and 0.5 % *DoLP* uncertainty for 440 nm, 550 nm, and 670 nm and 1.5% for 870 nm as inputs to GRASP. The 870 nm polarimetric data has larger uncertainty due to lower signal-to-noise ratio in the field data compared to 440 nm, 550 nm, and 670 nm, therefore, we give these data less relative weight in the retrievals.

 9. *Additional references for the standard reference frame used to calculate Q and U (Line #155)*

Thanks for pointing out this. This reference frame goes back to Chandrasekhar, Radiative transfer -1950. Additional references have been added.

The reference plane for the definition of $\mathbf{E}_\perp$ and $\mathbf{E}_\parallel$ is based on the local meridian plane, which is a standard reference frame used for reporting $Q$ and $U$ (Chandrasekhar, 1950; Emde et al., 2015; Hansen and Travis, 1974; Hovenier et al., 2004).

 10. *Add a hyper-link to the ACEPOL website (Line #163)*

Added a link to the ACEPOL campaign website.

The ACEPOL campaign (https://www-air.larc.nasa.gov/missions/acepol/index.html) was a collaborative effort of NASA and SRON (Netherlands Institute of Space Research) based out of Armstrong Flight Research Center (AFRC) in Palmdale, California, USA.

*11. Change the name of the instrument from 'airborne SPEX' to 'SPEX airborne' (Line #168)*

Corrected the name.

 airborne  (Smit et al., 2019),

*12. Mention the limitation of HSRL2 during ACEPOL (Line #186-189)*

For this study, aerosol extinction and AOD calculated using HSRL technique were only used. Whereas, for low aerosol loading cases these parameters are calculated using an assumption of lidar ratio of 40 sr. However, this limitation of HSRL2 has been added to the manuscript.

In some ACEPOL cases, due to atmospheric turbulence, interference in the HSRL2 measurement resulted in the inability to use the molecular channels at 355 nm and 532 nm to report the AOD, and required assumed lidar ratios of 20 sr and 40 sr over the ocean and land, respectively. However, for all the comparisons shown in this study, those cases were avoided and HSRL2 AOD reported here required no lidar ratio assumptions.

*13. Explain in more detail what is a kernel? and what are the parameters calculated for predefined optical and microphysical properties? (Line #219)*

According to the recommendation from the reviewer added an explanation on the GRASP kernel. Described the parameters in the kernel, and the range of the size and complex refractive index used for the generation of the single scattering kernel.

Relevant changes in the Section 3.2,

For the case of aerosol measurements, the forward model consists of a polarized Radiative Transfer (RT) code to calculate the radiance measured by the instrument and it  uses precalculated spheroidal kernel to calculate the contribution of single scattering by the aerosol particles following the strategy described by Dubovik et al. (2006, 2011). The kernel includes the pre-calculated full phase matrix elements, extinction and absorption for five log-normal size distributions with preselected size parameters for the range of real refractive index 1.33 to 1.7, and 0.0005 to 0.5 for the imaginary part of the refractive index for both spherical and non-spherical aerosol approximated by a mixture of spheroids with a fixed particle shape distribution derived in Dubovik et al., 2006. This approach allows for very fast and accurate calculations aerosol single-scattering properties in the wide range of refractive indices even for non-spherical aerosol. The details of the application of the kernels to satellite polarimetry are discussed in detail by Dubovik et al., 2011 (e.g. see Section 3.1, and Fig.4 in Dubovik et al., 2011).

*14. Do you use I, q, u or I and DoLP for the retrievals in this paper? (Line #230)*

For the demonstration of fit using individual pixels from the different land surfaces, $R_l$, q, u are used. For the AERONET and smoke case analysis on $R_l$ and DoLP are used for the GRASP fit. q and u calculated using the AirHARP is sensitive to the local meridian plane definition. Any error in the rotation of q and u will affect the retrieval and to avoid the extra source of uncertainty, $R_l$ and DoLP are used for the large scale retrievals. One exception is that for the ocean pixel study using the flight leg on 23-October-2017 T21:30 UTC we have used $R_l$, q, and u for the retrieval. This flight leg has been thoroughly quality checked for the error in the rotation of q and u from the instrument reference plane to the local meridional

plane. Throughout the manuscript, it has clearly mentioned which variables are used for the GRASP retrievals.

Changes implemented in Section 3.2,

For the AirHARP observations, $\mathbf{f}^*$ is a vector containing information of , $Q/I$,  $R_Q/R_I$), and $U/I$ (same as $R_U/R_I$) or  and $DoLP$ for all the spectral bands and viewing angles. GRASP is able to accept different configurations of the input parameters to make its retrieval. We will use the following sets of input in this work in different situations: ($R_I$, $Q/I$, $U/I$), or ($R_I$, $DoLP$). The text will explicitly state the inputs in each instance.

  15. *A well-known feature of GRASP is the multi-pixel retrieval capability, but it seems you are not using that. It would be good to mention this explicitly. (Line #240-243)*

Added a sentence stating that we haven't used the multi-pixel capability of GRASP in this study.

GRASP can perform retrievals using multi-pixel information in both spatial and temporal dimensions, however, in this study, we are not utilizing this feature due to the limited availability of data over the same place in the temporal dimension.

  16. *What is the relation between log-normal error distributions and the Levenberg-Marquardt method? (Line #247-249)*

Thanks to the reviewer for pointing this out. This sentence was not clear because of the phrasing. There is no relation between the LM method and log-normal error distribution. The sentence has been modified to make the idea clear. LM method is used as a solver for finding the solution of the non-linear minimization problem.

Relevant changes implemented in Section 3.2,

 In order to take into account, the non-negative character of measured and retrieved physical values in the retrieval optimization, the log-normal error distributions are assumed for all positively defined measured characteristics and the minimization is defined for logarithms of all positively defined retrieved parameters. The solutions to the set of equations in Eq. 3.1 are found by minimizing the term $\Psi(a)$ in Eq. 3.2. Since the radiative transfer in the atmosphere has pronounced non-linear character, the Levenberg-Marquardt (Levenberg, 1944; Marquardt, 1963) algorithm is harmoniously adapted into the statistically optimized fitting to assure the monotonic convergence of the iterative solution. These and other technical details of numerical inversion are described in Dubovik et al., 2011 and in-depth discussion of the above methodological aspects are also can be found in Dubovik and King, 2000, and Dubovik, 2006.

  17. *Add the information on the SPEX retrieved refractive index for a similar forest fire smoke from the ACEPOL campaign (Line #381)*

Added the information on the SPEX airborne retrieved refractive index value.

The values of RRI retrieved from AirHARP and shown in Figure 9(c) can be represented as a Gaussian distribution with a mode value of 1.55 for all wavelengths, while retrievals from the

RSP and SPEX airborne instruments during the ACEPOL campaign produced values of RRI of 1.56 and 1.58 respectively for a similar forest fire smoke (Fu et al., 2020).

> *18. Suggested to add 'and are close the values retrieved from SPEX airborne and RSP' to the sentence in Line #390*

As per the suggestion, the sentence has been modified to make the comparison more consistent.

Retrieved single scattering albedo values are well within the range measured during the FLAME 2 experiment using a photo acoustic spectrometer and a nephelometer and are close to the retrieved values from SPEX airborne and RSP by Fu et al., 2020.

> *19. Comment on Line #400-401 'Actually, those retrievals are for a smoke plume on 9 November. There, the % sphere seems to agree quite well with the HSRL2 depolarization ratio. Could you compare that here as well?'*

A comparison with the HSRL2 aerosol depolarization ratio and the GRASP retrieved spherical fraction (SF) as a function of along-track flight pixels is plotted in Figure AR1.1. It also includes the trend of Angstrom Exponent (AE) along with the $AOD_{532\ nm}$ measured by HSRL2. From the AE value, it is clear that we have fine mode dominated particle size distribution and, correspondingly, the sensitivity of SF in the aerosol retrieval is low compared to the situation when the coarse mode aerosol, such as dust, dominates. The range of values we see for the aerosol depolarization from HSRL2 suggests that particles are fine but may have some non-spherical shapes. The HSRL2 data also suggest that the depolarization is decreasing when we go closer to the smoke plume. In these regards, HARP retrievals do not show the corresponding decrease of SF, however, the effect of particle non-sphericity on the radiances and polarization observed by passive instruments is rather small for fine mode dominated aerosols (e.g. see analysis by Dubovik et al., 2006) and the retrieval of SF has very limited accuracy. A detailed study on how the aerosol depolarization ratio change for spherical and non-spherical particles for a fine-mode dominated aerosol will help us understand better about this problem.

[Figure]

Figure AR1.1 (a) Columnar aerosol depolarization ratio and aerosol optical depth (AOD) at 532 nm from HSRL2 measurement, spherical fraction and angstrom exponent (AE) (using 440 and 870 nm bands) exponent retrieved using GRASP and AirHARP observation for the forest fire smoke plume on 9[th] November 2017 T19:30 UTC as a function of the along-track flight pixels. (b) Scatterplot of collocated columnar aerosol depolarization ratio and mean SF values

[Figure]

Figure AR1.2 (a) Columnar backscatter weighted angstrom exponent (AE) calculated using 355 and 532 nm observations and aerosol optical depth (AOD) at 532 nm from HSRL2 measurement, spherical fraction and angstrom exponent (AE) (using 440 and 870 nm bands) exponent retrieved using GRASP and AirHARP observation for the forest fire smoke plume on 9th November 2017 T19:30 UTC as a function of the along-track flight pixels. (b) Scatterplot of collocated columnar AE from HSRL and AE value from AirHARP.

Figure. AR1.2 shows that there is a clear correlation between the AE measured by HSRL2 and AirHARP at particularly AOD(532 nm) > 0.4. This implies that the particle size distribution changes with the location of the plume. The corresponding author is currently working on a project to analyze the retrieved aerosol and surface product from AirHARP and SPEX airborne polarimeters to understand how the synergy between two polarimeters can benefit the PACE mission. As a part of future study, this concern by the reviewer will be analyzed in detail. Given the results of this analysis and the other paper in process, the authors choose to not compare columnar aerosol depolarization ratio and SF in the current paper.

> 20. Comment on Line #432-433 'It would be interesting to also show results for the microphysical properties for this case as they can be directly compared to the values of Fu et al (2020) for SPEX airborne and RSP. Also, the sphericity can be checked using the depolarization ratio'

Results of the 9th November 2017 T19:31 UTC plume is not added to the manuscript because of the non-homogeneity of the plume compared to 27th October 2017 T 18:15 UTC one. The retrievals of aerosol properties show a high variability along the smoke plume source and edges and this can bias the mean calculation of aerosol and microphysical properties reported for this case. Further studies are ongoing based on the flight leg on 9th November 2017 T19:31 UTC to improve the retrievals and understand how the optical property of smoke aerosol changes with the age/location from the plume source. We have added a sentence in the text to explain why particle properties will not be shown. A list of the mean aerosol optical properties for the same scene is tabulated in Table AR1.1 for your reference. As mentioned in the reply for comment #19, the corresponding author is working on a project to compare the aerosol and surface products from two polarimeters onboard on ACEPOL.

Table AR1.1: Mean aerosol optical and microphysical properties retrieved for the smoke scene on 9th November 2017 T19:31 UTC (for pixels with AOD$_{440 nm}$ > 0.4).

| Spectral Band | Single Scattering Albedo | Spherical Fraction (%) | Angstrom Exponent# | Real Refractive Index (RRI) | Imaginary Refractive Index (RRI) |
|---|---|---|---|---|---|

| | | | | | |
|---|---|---|---|---|---|
| 440 nm | 0.83±0.1 | | | | 0.034±0.02 |
| 550 nm | 0.84±0.1 | 70±34%@ | 1.58±0.36 | 1.54±0.03 | 0.028±0.02 |
| 670 nm | 0.80±0.13 | | | | 0.035±0.02 |
| 870 nm | 0.76±0.15 | | | | 0.039±0.02 |

**Angstrom Exponent calculated using the AOD at wavelength bands 440 nm and 870 nm of the AirHARP**

@Retrieved spherical fraction includes a significant number of pixels with SF ~ 99%

*21. Does the agreement improve when using a stricter chi2 filter? (Line #448)*

Yes, the agreement improves when stricter $\chi^2_{norm}$ filtering $(\chi^2_{norm} > 5)$ is used to remove bad fits. The comparison is plotted in Figure AR1.3. The bias and MAE improve over the less strict $\chi^2_{norm}$ filtering. However, several higher AOD points are missing from the analysis.

[Figure]

Figure AR1.3 Scatterplot of the collocated observation AOD from HSRL2. The plot is the same as Fig. 13 (b) in the manuscript but a stricter $\chi^2_{norm} < 5$ filter is used for the removal of pixels. Also, if the box used for the mean AOD doesn't have enough pixels to do the statistics.

*22. I think the 3D effect is only important very close to the source but not for the majority of pixels here. In Fu et al (2020) a stricter chi2 criterion for SPEX airborne and RSP was used and still, some high AOD values were in the comparison. Still, the spatial mismatch between airHARP and HSRL2 is probably an important reason for disagreement and I think this is much more relevant than the 3D effects (for most pixels). (Line #457-458)*

Thanks for pointing this out. This mismatch is mainly caused by the difference in the spatial resolution of the measurement, which will affect the comparison over the plume edges and boundaries where there is a sharp change in the AOD values. Also, this will affect the pixels near to the plume source, since each viewing angle of the instrument will be looking at

different thickness and this violates the plane parallel assumption. To accommodate this issue, we will have to consider different thickness for different viewing angles in 1D RT code or will have to use 3D RT code completely in the forward model calculations. The sentence has been modified to avoid the confusion.

Matching the HSRL2 AOD in regions of heterogeneity is challenging due to spatial mismatch between AirHARP and HSRL2 pixels. This will create issues where there is a sharp variation in the AOD, like close to the source, and in the boundary of the smoke plume. The different cross-track pixel size between the HSRL2 and AirHARP measurement makes the intercomparison difficult to interpret in some cases. For points near the plume source, higher pixel variability may also bias AirHARP AOD retrievals performed at the same general location as the HSRL2 measurement. In a scene with this much complexity there is additional uncertainty in matching multiangle views for the AirHARP retrieval, because each viewing angle of the instrument will be looking at a different plume thickness and this violates the plane parallel assumption.

> *23. A reference to Stap et al. would fit here: Atmos. Meas. Tech., 8, 1287–1301, 2015*
>     *www.atmos-meas-tech.net/8/1287/2015/doi:10.5194/amt-8-1287-2015 (Line #469)*

Added the reference in Section 5.2,

To further protect the algorithm from subpixel inhomogeneity and other features inappropriate for retrieval a $\chi^2_{norm}$ ¡ 1.5 filter is used to remove the bad pixels/fits which may be caused by the presence of thin clouds (Stap et al., 2015) or due to the inability of surface reflection models to represent the directional reflectance from a complicated surface.

> *24. The 2E-3 degree seems to contradict with the 5.5x5.5km2 area where airHARP AOD is averaged. (Line #470-474)*

Actually, 2e-3 degree is used to locate the central AERONET pixel (a single-pixel) and used a 10x10 pixel square around the AERONET station pixel to calculate mean AOD.

Changes in Section 5.2,

To collocate the AERONET station (a single-pixel ) within the AirHARP image, the latitude and longitude of the AERONET location are matched to the AirHARP latitude and longitude with a tolerance of $2\times10^{-3}$ degree, which is approximately equivalent to 200m on the ground. An area of 5.5 km x 5.5 km (10x10 retrieval pixels) around this collocated pixel is used for the calculation of the area mean AOD from the AirHARP retrievals, and this is matched to a one-hour temporal mean from AERONET measurements.

> *25. Question on how the AOD from AERONET measurements is interpolated to AirHARP bands? (Line #492-493)*

AOD is interpolated linearly in log-log space using the Angstrom Exponent calculated using the closest bands around the wavelength of interest.

Added a sentence in Section 5.2,

AERONET measured AOD are interpolated linearly in log-log space using the AE to AirHARP spectral bands for 1:1 comparison.

*26. If you choose to mention this then for completeness you should also mention that airMSPI and SPEX airborne have smaller biases against AERONET (Line #500-501)*

The authors were talking about the negative bias we saw in the AOD range of 0 to 0.1. In the Fu et. al 2020, for the AEROENT AOD comparison, SPEX airborne and AirMSPI were showing a positive bias in this low AOD range mentioned. Whereas RSP shows the negative bias for this AOD range. But this sentence in the manuscript has been removed after the new AERONET comparison using Ross-Li based land model

Removed the sentence in Section 5.2,

*27. Mention in the conclusion also the different retrieval setups used for the different cases. (Line #514)*

Findings based on the two approaches have been added to the conclusion.

In situations with low aerosol loading (AOD < 0.17) over land, a simplified retrieval approach based on GRASP/Models kernel approximating aerosol as an external mixture of five aerosol components is also used for the AOD retrievals. One advantage of using this simplified kernel is that it retrieves a significantly smaller number of aerosol parameters compared to the standard GRASP/Five mode kernel and performed well for low aerosol loading cases in an AERONET comparison, despite the simplifying assumption of prescribed complex refractive index for each aerosol component. AOD retrieved from AirHARP using GRASP, matches collocated AERONET observations to within +0.018/-0.04 with a minimum MAE of 0.01 in 440 and 550 nm bands and maximum of 0.015 in the 870 nm spectral band.

*28. This part should be left out. The performance for 7 AERONET overpasses is in no way representative for global performance and should hence not be compared to the quoted uncertainties for MODIS. Even not with the reservation that you make at the end. (Line #530-534)*

We feel that it is good to put the numbers that come out of the collocation with AERONET into some kind of perspective for people who are novices to MAPs can digest. Yes, of course, it is a microscopic data sample compared to global statistics, but the caveat is stated clearly. If the reviewer has a suggestion on how to put the results into context, the authors are open to new wording. Otherwise, the authors prefer to keep the statement, as is.

*29. This suggests that you validated these properties. I would weaken it a bit, e.g. "...that we retrieved with GRASP optical and microphysical ..." (Line #537)*

The sentence has been modified to avoid confusion.

It is in one of these situations that  optical and microphysical characteristics of the smoke, in addition to AOD, are retrieved using the GRASP software.

*30. See my comments above. I do not believe that the 3D RT effect is the problem for most of the pixels (it only is very close to the source) but rather the spatial mismatch between airHARP and HSRL2. On the other hand, it could even be related to HSRL2 itself. Also, SPEX airborne and airHARP see a similar trend in underestimation while they all have different spatial samplings ... (Line #546-548)*

Agree with the reviewer that this is mainly caused by the spatial mismatch between the HSRL2 and AirHARP pixels. But the 3D structure also plays a role in the mismatch near the smoke source. Also, it can be due to the HSRL2 itself. The sentence was modified.

Note that when the plume is highly heterogeneous, the smaller cross-track footprint of HSRL2 measurements, relative to AirHARP makes collocation extremely difficult. Also, the complex structure of the plume  violates the plane-parallel aerosol layer assumption of the retrieval, adding uncertainty and bias.

> 31. *This is still to be demonstrated for the microphysical properties and SSA, so please weaken the statement a bit. (Line #553-554)*

The sentence has been modified to make it weaker and clearer.

Thus, the encouraging results demonstrated here show that when combined with GRASP inversion methods, HARP measurements have the potential to be used to retrieve  accurate measures of AOD. With sufficient aerosol loading and homogeneity, aerosol optical/microphysical characteristics can be retrieved over a broad area.

> 32. *It would be nice to stress the synergy with SPEXone (https://www.sciencedirect.com/science/article/pii/S0022407318308653), the other polarimeter on PACE. (Line #559)*

A description of the synergy between the three instruments onboard on the PACE mission is provided with references.

Furthermore, HARP has the potential to provide new characterization for clouds (McBride et al., 2020) and surface properties over various surface types, land and ocean. The PACE mission also opens the corridor for a synergetic observation using the Ocean Color Instrument (OCI) along with the two multi-angle polarimeters: HARP2 and SPEXone. OCI is a hyperspectral, wide swath radiometer, HARP2 — a wide swath multi-angle polarimeter and SPEXone — a hyperspectral narrow swath multi-angle polarimeter. The combined spectral and spatial coverage and resolution of these three instruments will provide an unprecedented dataset for the atmospheric, ocean and terrestrial science research communities (Frouin et al., 2019; Hasekamp et al., 2019; Remer et al., 2019).

> 33. *Please also acknowledge the Dutch funding of ACEPOL flight hours: "The Dutch contribution to the ACEPOL flight campaign hours was funded by SRON Netherlands Institute for SPACE Research and by NSO/NWO under project number ALW-GO/16-09. Same for the NASA flight hours (We acknowledge the former Aerosol, Cloud, Ecosystem (ACE) program at NASA's Earth Science Division as a sponsor for ACEPOL flights.) (Line #635)*

Added a sentence in the acknowledgement,

We acknowledge the former Aerosol, Cloud, Ecosystem (ACE) program at NASA's Earth Science Division as a sponsor for ACEPOL flights. The Dutch contribution to the ACEPOL flight campaign hours was funded by SRON Netherlands Institute for SPACE Research and by NSO/NWO under project number ALW-GO/16-09.

**Other comments**

*34. It seems that 3 different GRASP setups are used for 3 different cases:*

*- GRASP for fixed aerosol models, together with the RPV surface model for the AERONET comparison.*

*- Retrievals for 5 aerosol modes with fixed refractive index, together with the Ross-Li surface model for the Rosamond Dry lake.*

*- 'Full" retrievals for 5 aerosol modes where also microphysical properties are retrieved for the smoke case. Here also the Ross-Li surface model has been used.*

*These 3 setups should be clearly described in the Theory section and not in the Result sections where they are being used. Also, the choice for a given setup should be better motivated, and perhaps also some numbers for retrievals in other setups should be quoted.*

*If possible, I would even suggest restricting to 2 setups: the ones corresponding to 'full' 'and models', and using the same surface model for all over land cases.*

Authors would like to thank Otto for suggesting this comment and as per the recommendation of the reviewers, only one land surface model (Ross-Li based BRDF) is now used in the retrievals over the land pixels. The updated manuscript includes a comparison of the two different retrieval methods. Validation plots of AirHARP vs AERONET AOD for the two methods: a) five fixed modal log-normal size distribution GRASP kernel (GRASP/Five mode) and b) aerosol component-based GRASP kernel (GRASP/Models) where complex refractive index, spherical fraction and aerosol distribution are fixed. Only the weight for each aerosol type can change in the retrieval. Case (b) has a significant reduction in the number of retrieved aerosol parameters from 15 to 6, which will reduce the non-linearity of the minimization problem. This simplified method seems to perform well over the five modal log-normal approach for these low aerosol loading cases.

In section 5.2 added a discussion of the two aerosol GRASP kernel approaches and the results,

[revised manuscript text omitted]

---

## Author Comment (AC2) · 23 Jun 2020

**Reply to Reviewer #2 comments**

The authors would like to thank anonymous reviewer #2 for taking the time to review this manuscript and providing valuable comments and recommendations on an open discussion. For the ease of cross reading, authors responses are given under the reproduced version of each comment by reviewer #2. Each of the comments from the reviewer are numbered and the changes made in the manuscript are shown using a red font under the authors response given in blue font.

**Comments**

1. *The authors review the airborne polarimeters (SPEX, RSP, AirMSPI). It is also necessary to review aerosol retrieval algorithms used by these sensors and published elsewhere.*

References to the current retrieval algorithms used by different sensors have been added to the revised manuscript. Also redirecting the readers to more comprehensive review papers on polarimetric aerosol sensing and retrieval algorithms.

Changes implemented in Section 1,

There are several aerosol retrieval algorithms specifically optimized for MAPs which includes: SRON multi-mode inversion algorithm for SPEX airborne (Fu et al., 2020; Fu and Hasekamp, 2018); Microphysical Aerosol Property from Polarimeters (MAPP) (Stamnes et al., 2018) and GISS/RSP algorithm (Knobelspiesse et al., 2011; Waquet et al., 2009) for RSP; correlated multi-pixel and joint retrieval algorithm for AirMSPI developed at Jet Propulsion Laboratory (JPL) (Xu et al., 2017, 2019). This list is not complete, for a comprehensive review of the polarimetric remote sensing of atmospheric aerosols based on MAPs, we encourage the readers to refer to several reviews in the literature (Dubovik et al., 2019; Kokhanovsky et al., 2015; Remer et al., 2019).

2. *Associated with the comparison of AirHARP AOD results against AERONET reference data in Figure 14, how do the comparison of single scattering albedo and non-spherical particle fraction look like? Though the AOD loading is low, are we still able to see a trend of improved agreement of these properties as AOD increases?*

Since the aerosol loading is very low, most of the AERONET stations do not have SSA retrievals for many of the collocated observations. For those AERONET stations that do have a retrieval for SSA are only at Level1.5, which are not quality assured. There are 6 collocated observations for the SSA. A comparison plot for the GRASP retrieved SSA using Air-HARP observation and L1.5 SSA retrieved using AERONET almucantar measurements is given in Fig. AR2.1. The figure also includes the plot which shows the trend of difference in SSA with an increase in the AOD measured by the AERONET. The same plots are given for two cases: a) five modal log-normal size distribution based GRASP/Five mode kernel and b) aerosol component-based GRASP/Models kernel where complex refractive index, spherical fraction and aerosol distribution are fixed. Only the weight for each aerosol components can change in the retrieval. There is a significant reduction in the number of retrieved aerosol parameters from 15 to 6 in case (b), which will reduce the non-linearity of the minimization problem. In the Reply to Reviewer #1, we show an analysis that demonstrates that the

sensitivity to the spherical fraction is small for fine mode dominated particle size distribution and we can't expect a good agreement with the AERONET retrieved SF.

The point is that there is insufficient aerosol loading with the AERONET match-ups to expect a good retrievals of particle properties from any inversion. This is why AERONET has no Level 2 Quality Assured retrievals, and why we do not want to include any comparisons in the paper.

[Figure]

Figure AR2.1: (a) and (b) are the scatter plots of retrieved SSA from AERONET and AirHARP observations. For the case of (a), AirHARP retrievals use a GRASP five modal log-normal kernel, where the complex refractive index and spherical fraction are retrieved along with the weight for each lognormal mode in the PSD (See Table. 3 in the manuscript for more details on the fixed size modes). On the other hand, case (b) uses an aerosol type-based GRASP kernel where optical properties are precomputed using different aerosol types mentioned in Table. 6. Subplots (c) and (d) show the trend of difference in SSA retrieved from AERONET and AirHARP observations with an increase in AOD measured by AERONET stations. Plots (c) and (d) are based on the kernels mentioned in (a) and (b) respectively.

3. The authors need to be clearer on the adoption of size components in AirHARP retrieval. Table 3 and Table 6 give two different size components in retrieval. If I

*understand well, size components in Table 3 are the default option in GRASP while size components in Table 6 were created for AirHARP retrieval. What is the difference in retrieval based on these two different assumptions? Which one gives a better fit to AERONET AOD and SSA? As the authors pointed out "This simplified approach significantly drops the complexity of the aerosol model by reducing the number of parameters retrieved in the joint retrieval. It helps in reducing the nonlinearity of the inverse problem and makes the separation of the surface and aerosol signal much less complicated compared to a five-lognormal mode kernel." However, would it also possible that the size components in Table 6 (with a fixed refractive index for each mode) also increase the risk of getting more subjection to pre-determined aerosol modeling errors? If so, how large is the modeling error?*

Table 3 is the default option used for the retrieval in this paper. It is not the default option in GRASP. GRASP offers many options. Its most generalized version of a kernel has 22 bins for the particle size distribution plus all of the other particle properties. But to reduce the complexity for the AirHARP retrievals, based on the experience of PARASOL/GRASP retrievals, we are using a different option with the predefined five modal log-normal distribution mentioned in Table 3. This one has 15 free parameters including spectrally dependent complex refractive index, spherical fraction, etc. For the AERONET comparison, we invoke yet a different GRASP option. We used the GRASP/Models approach representing aerosol as an external mixture of predetermined aerosol components, where the number of free parameters is further reduced to 6 and the complex refractive index and the spherical fraction are fixed for each aerosol type. Only the concentration (weight) of each aerosol component in Table 4 (old Table 6) are retrieved during the inversion. Since the aerosol loading for the AERONET comparison cases is below 0.17, this approach seems to be working well compared to the one mentioned in Table 3 (complex refractive index and spherical fraction are retrieved parameters).

We have worked hard to try and clarify the text in this matter, and have added a plot (Figure 14 in the revised manuscript) that compares the 15 and 6 free parameter options.

Figure AR2.1 gives us an insight into modeling error because of the assumption of the aerosol components. For this range of aerosol loading, the assumption of aerosol type will not significantly affect the $R_i$, and *DoLP* calculations. However, this assumption may not hold for higher aerosol loading where a predetermined aerosol model will create modeling error in $R_i$ and *DoLP* calculated using the forward model.

Changes implemented in the manuscript relevant to this comment:

In section 3.2 describing the different GRASP kernels and surface models used in the retrievals,

[revised manuscript text omitted]

Change implemented in the conclusion based on the new results,

In situations with low aerosol loading (AOD ¡ 0.17) over land, a simplified retrieval approach based on GRASP/Models kernel approximating aerosol as an external mixture of five aerosol components is also used for the AOD retrievals. One advantage of using this simplified kernel is that it retrieves a significantly smaller number of aerosol parameters compared to the standard GRASP/Five mode kernel and performed well for low aerosol loading cases in an AERONET comparison, despite the simplifying assumption of prescribed complex refractive index for each aerosol component. AOD retrieved from AirHARP using GRASP, matches collocated AERONET observations to within +0.018/-0.04 with a minimum MAE of 0.01 in 440 and 550 nm bands and maximum of 0.015 in the 870 nm spectral band. Thus, we note an overall low bias of -0.002 to -0.009, depending on wavelength..

4. *As the AOD loading during the ACEPOL field campaign is low, it provides a very good testbed for surface retrieval. Is there any comparison of surface BRDF and pBRDF as retrieved from AirHARP and other sensors such as SPEX, RSP, and AirMSPI?*

The authors agree with the reviewer that this dataset/campaign is a good testbed for surface retrieval. This is an ongoing project and will be part of a publication expected from that project. For now, the authors believe that it is beyond the scope of this paper.

**References**

[revised manuscript text omitted]